

# Bounded and categorized: targeting data assimilation for sea ice fractional coverage and non-negative quantities in a single column multi-category sea ice model

Molly M. Wieringa[1], Christopher Riedel[2], Jeffrey L. Anderson[2], and Cecilia M. Bitz[1]

[1]University of Washington, Department of Atmospheric Sciences, Seattle, WA, USA
[2]National Center for Atmospheric Research, Data Assimilation Research Section, Boulder, CO, USA

**Correspondence:** Molly Wieringa (mmw906@uw.edu)

**Abstract.** A rigorous exploration of the sea ice data assimilation (DA) problem using a framework specifically developed for rapid, interpretable hypothesis testing is presented. In many applications, DA is implemented to constrain a modeled estimate of a state with observations. The sea ice DA application is complicated by the wide range of spatio-temporal scales over which key sea ice variables evolve, a variety of physical bounds on those variables, and the particular construction of modern complex sea ice models. By coupling a single-column sea ice model (Icepack) to the Data Assimilation Research Testbed (DART), the grid-cell response of a complex sea ice model is explored with a range of ensemble Kalman DA methods designed to address the aforementioned complications. The impact on the modeled ice-thickness distribution and the bounded nature of both state and prognostic variables in the sea ice model are of particular interest, as these problems are under-examined. Explicitly respecting boundedness has little effect in the winter months, but correctly accounts for the bounded nature of the observations, particularly in the summer months when prescribed SIC error is large. Assimilating observations representing each of the individual modeled sea ice thickness categories consistently improves the analyses across multiple diagnostic variables and sea ice mean states. These results elucidate many of the positive and negative results of previous sea ice data assimilation studies, highlight the many counter-intuitive aspects of this particular data assimilation application, and motivate better future sea ice analysis products.

## 1 Introduction

Recent rapid Arctic change has emphasized the influence of sea ice on the global climate system, our incomplete understanding of its recent history, and many shortcomings of current sea ice models. The tide of interest in addressing these issues is well-reflected in the accelerating application of data assimilation techniques in both sea ice reconstruction projects (Schweiger et al., 2011; Sakov et al., 2012; Mu et al., 2018; Williams et al., 2022) and modelling studies (Zhang et al., 2021). Data assimilation, or DA, is a set of objective methods through which observations of a system are blended with a modeled estimate of that system. Through this blending, DA injects the real-world information gained via the observations, which are typically limited in space and can be intermittent in time, into a model capable of integrating that information forward in a spatio-temporally continuous, physically realistic manner. DA is most commonly used to obtain accurate initial conditions for numerical weather



prediction models, but can also be deployed in climate studies to reconstruct unobserved variables by synchronizing observable
components of a system with nature or to infer the correct parameterization values that should be used in earth system models.
To date, most sea ice DA applications have employed ensemble Kalman filtering (EnKF) methods, a family of DA algorithms
based on the Kalman filter (Kalman, 1960; Evensen, 2003; Houtekamer & Zhang, 2016). EnKF methods approximate the
application of a true Kalman filter by sampling the system of interest using model ensembles. In practical applications, the
adjustments made by these filters can be considered in four steps. Firstly, the model is used to generate an ensemble of
forecasts. Secondly, a version of the Kalman filter is applied to update the model's estimates of the observed quantity. Here,
this will be referred to as observation-space incrementing. Thirdly, the adjustments made in observation space are used to
determine the corresponding updates applied to the variables comprising the model state. This step is hereafter referred to as
state-space regression. Together, observation-space incrementing and state-space regression are collectively known as filtering.
Lastly, the updated model state is used to initialize the next forecast step. All together, this process is termed a DA cycle.

Substantial nuance can arise in the cycling process depending on the characteristics of the system in question. This makes
DA in any earth system component model an intricate undertaking, and one often specifically tailored to the problem at
hand. For sea ice this is particularly true, as sea ice models and observables unite many distinct challenges for DA in one
system. First, similar to atmospheric variables such as cloud fraction, sea ice variables tend to be bounded. For example, ice
cannot be negatively thick; the fraction of a model grid cell covered by ice (sea ice concentration) cannot fall below zero or
exceed one. The Kalman methods applied to sea ice problems are based on assumptions that the model ensemble and the
observation error distribution are normal distributions, which thereby linearizes the filtering process. For system variables that
are bounded, however, the use of normal distributions in the filtering algorithm can produce adjustments during observation-
space incrementing that violate physical bounds. When these violations are corrected (typically through a postprocessing step),
the model ensemble mean is artificially shifted away from the bound, leading to analysis inaccuracies. While non-Gaussian
ensemble DA methods that avoid the use of normal distributions have been proposed, their application in high-dimensional
systems has been limited (Riedel & Anderson, 2023; Anderson, 2010).

    Secondly, the relationship between variables observed in the real world and modeled in the sea ice state is not straightforward.
Sea ice observing systems measure variables such as sea ice concentration (SIC) or sea ice thickness (SIT). However, SIC
and SIT are *diagnostic* in modern sea ice models, which typically evolve through an ice-thickness distribution (ITD). The
ITD parameterizes sub-grid scale thermodynamic processes that are strongly dependent on ice thickness (Bitz & Roe, 2004;
Chevallier & Salas y Melia, 2012) by expressing the distribution of ice variables in a grid cell as functions of the ice thickness.
In practice, the ITD describes a range of thicknesses within each grid cell and discretizes that range into an arbitrary number of
thickness categories. Sea ice area and volume (and the snow volume atop the sea ice) are then similarly distributed across the
thickness categories (Thorndike et al., 1975) and the evolutionary equations of the sea ice model are applied to each category
individually. Observed SIC, SIT, and snow depth (SND) are aggregates of the "categorized" model variables of ice area ($A_{\mathrm{ice},n}$),
ice volume ($V_{\mathrm{ice},n}$), and snow volume ($V_{\mathrm{sno},n}$), respectively; the latter three sets of variables represent the sea ice state. Thus,
while SIC and SIT are updated during observation-space incrementing when SIC or SIT observations are assimilated, the
updates to the aggregate values are regressed out to each of the categorized variables during the state-space regression. The



diagnostic SIC and SIT output at the end of the process are then re-aggregated, updating categorized state variables; their
accuracy relies not only on the direct filter updates on the aggregate quantities, but also on the model ensemble's relationship
between the aggregated quantities and each of the categorized variables in the model state. Few studies have presented the
impact of assimilating SIC or SIT on each of the model's categories individually, which raises the question of how well the
process and impact of assimilating any observation into distribution-based sea ice models is understood. Recent work by
Williams et al. (2022) documents the first attempt to assimilate an "observed" ice thickness distribution rather than just an
aggregate observation into the sea ice component of a global climate model, with mixed results.

Both the non-Gaussian, bounded nature of sea ice and the relationship between aggregate observables and categorized state
variables likely have important ramifications for sea ice DA but remain under-explored. This study presents a single-column sea
ice data assimilation framework that allows for rapid hypothesis testing while retaining the thermodynamic physics and ITD of
a complex sea ice model. Within this idealized framework, the impact of using DA algorithms that respect the boundedness of
sea ice model variables and observations is explored, as is the ITD response of the model when assimilating aggregate versus
categorized area and thickness observations. Section 2 provides an overview of the data assimilation framework and experimen-
tal methodology; Section 3 presents a discussion of the results generated by a suite of DA experiments targeting boundedness
and categorized observations; Section 4 contextualizes this work with respect to more practical sea ice DA applications; Section
5 concludes.

## 2   Model and Methods

The data assimilation framework used in this study couples the Data Assimilation Research Testbed (DART, Anderson et al.,
2009) to Icepack (Icepack, 2020), a single-column version of the CICE sea ice model; the latter is widely used as the sea ice
component of several Earth system models and in stand-alone sea ice studies. In keeping with naming conventions developed
in coincident work (Riedel et al., 2023), this framework is referred to as CICE-SCM-DART. A sea ice quantity produced by a
CICE-SCM-DART experiment is hereafter differentiated from the corresponding synthetic observations assimilated using the
terms "modeled" and "observed", respectively.

### 2.1   Icepack

Icepack is maintained as the column physics module of CICE, with consistent thermodynamics, mechanical redistribution,
and tracer support. The model evolves conditions in four independent grid cells, each with a different surface type (land, open
ocean, slab ice, or a categorized ITD). These four grid cells do not communicate in any way; as such, only output from the ITD
grid cell is retained.

For use in the CICE-SCM-DART framework, 30 instances of Icepack are forced by unique atmospheric conditions extracted
from randomly selected members of the CAM6 + DART reanalysis product (Raeder et al., 2021). The sea ice model is tuned
to the atmospheric forcing by setting the snow grain radius parameter ($R\_snw$) to a value of $-2$. This choice prevents dis-
continuous behavior in ice concentration related to ice-albedo feedback during the melt season and thus allows for a wider





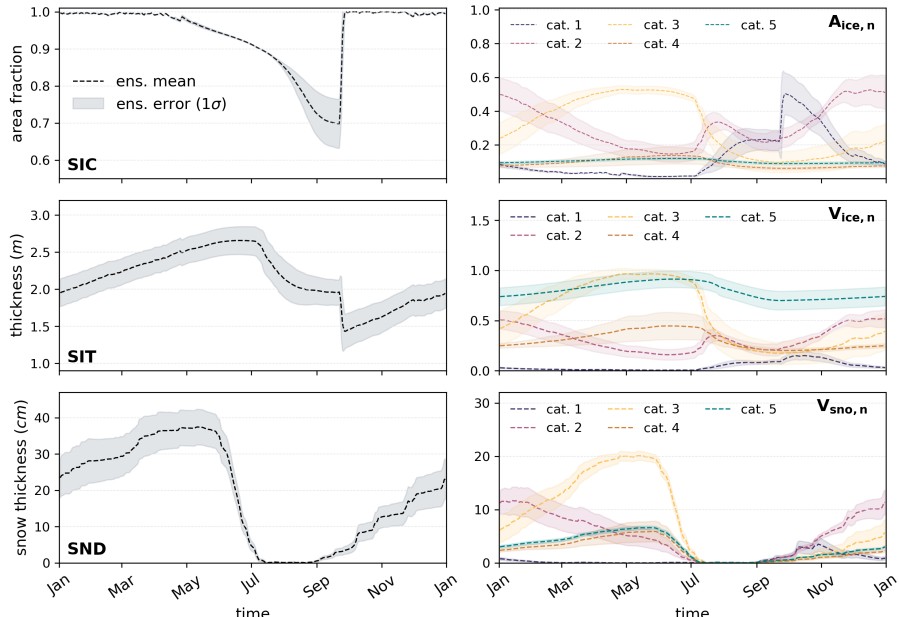

**Figure 1. The FREE ensemble's three aggregate variables and the state variables from which they are derived.** The aggregate variables shown on the left (SIC, SIT, and SND) are sums of the categorized state variables (category ice area, ice volume, and snow volume on the right. There are five thickness categories, where category 1 refers to the thinnest ice (0-0.64m) and category 5, the thickest ice (4.57m and thicker). Dark lines indicate the ensemble mean of each variable and lighter shading represents the ensemble standard deviation around the mean.

summertime ensemble spread. The number of categories used in the ITD is set to 5. All other sea ice model parameters are held at their default values. Each instance of Icepack is also coupled to a slab ocean; the ocean initial conditions and heat flux convergence forcing are consistent across the 30 members and are derived from the ocean component output of a fully-coupled historical simulation from the Community Earth System Model (CESM2). Both the ocean and atmosphere data sets represent grid cells nearest $75.54^oN$, $174.45^oE$, a point that straddles the East Siberian and Chuckchi Seas and experiences seasonal sea-ice advance and retreat.

The ensemble is spun up over a 10-year period during which the atmospheric conditions cycle continuously over the year 2011. No assimilation occurs during this period. Once spin-up is complete, a final year-long ensemble simulation is produced as a control case for the assimilation experiments. This simulation, which is also absent any assimilation, is hereafter referred to as the FREE case and is outlined in Fig. 1. Both categorized state variables (right) and their diagnosed aggregates (left) are shown, as both can be observed and adjusted by assimilation.



## 2.2 DART

DART is a modular data assimilation framework developed by the Data Assimilation Research Section at the National Center for Atmospheric Research. DART interfaces with many models that range in complexity from the Lorenz 3-variable chaotic model to the Community Atmosphere Model (CAM6). DART implements the four-step cycling approach outlined in the introduction: forecast, observation-space incrementing, state-space regression, and re-initialization. DART currently includes 10 filtering algorithms, including variants on the ensemble Kalman filter and several kernel and particle filter options. The default filter, the Ensemble Kalman Adjustment Filter (EAKF; Anderson 2001), implements a square-root filtering approach that increases the stability and efficiency of assimilating with smaller ensemble sizes. Like most traditional ensemble filtering approaches, the EAKF makes Gaussian assumptions for the model ensemble and the observation error distributions.

Recently, Anderson (2022) developed a novel filtering approach known as the quantile-conserving ensemble filtering framework (QCEFF). QCEFF alters the process by which the updated ensemble is sampled from the analytical blend of the model ensemble distribution and the observation error distribution. As a result, DART users can prescribe non-Gaussian distributions that may better represent the model ensemble or observation of interest. For example, in the sea ice problem, QCEFF allows the user to prescribe distributions that respect sea ice bounds, a level of detail that cannot be attained by EAKF or other Gaussian filters. In this framework the user can prescribe a distribution for each observable or state variable, as well as differentiate the distribution used for observation-space incrementing versus state-space regression; this kind of choice allows the user to tailor the DA framework to the problem at hand in every step of the filtering process. When the user prescribes normal distributions in the QCEFF framework, the solution collapses to the EAKF.

We employ QCEFF to examine whether explicitly accounting for sea ice boundedness can improve sea ice assimilation analyses. To do so, we compare four different filtering approaches, outlined in Table 1. These filtering approaches use varying combinations of normal and piece-wise rank histogram distributions in the observation-space incrementing and state-space regression steps of the filter. Piece-wise rank histogram distributions prescribe no more information about the distribution of the sea ice system than can be gained from the discrete ensemble members themselves and can capture physical bounds; their use in DART's filtering algorithms and for sea ice applications is discussed in more detail in (Anderson, 2020), (Riedel & Anderson, 2023) and (Riedel et al., 2023). The use of bounded normal rank histogram (BNRH) distributions in state-space regression is addressed in (Anderson, 2023).

## 2.3 Experimental Setup

All experiments performed for this study follow a perfect-model Observing System Simulation Experiment (OSSE) protocol ((Zhang et al., 2018; Riedel & Anderson, 2023; Riedel et al., 2023)), a methodology typically used to identify the impact of assimilating a set of proposed or synthetic observations. The use of synthetic observations allows for a close inspection of DA filter performance given a set of observations derived from a known state. Here, several different kinds of synthetic sea ice observations are assimilated using each of the filter types listed in Table 1. Each experiment was branched from the end of





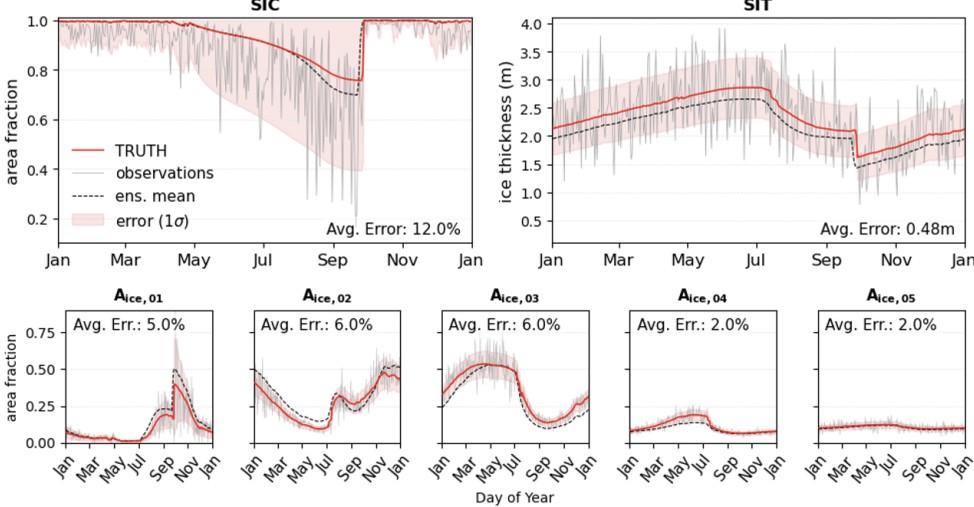

**Figure 2. Synthetic observations extracted from a randomly selected member of the FREE ensemble.** The observation assimilated are shown in grey lines for SIC (top left), SIT (top right) and category ice area ($A_{ice,01}$-$A_{ice,05}$) on the bottom row. The TRUTH from which the observations are generated in shown in the solid red line, while the FREE ensemble mean is shown in the dashed black line. The observation error standard deviation ($1\sigma$) is shown as red shading around TRUTH.

the ensemble spin-up period, assimilated observations for a year, and was then compared to the FREE case. The assimilation

experiments presented in the results are listed in Table 2.

The observations assimilated (a subset of which are presented in Fig. 2) are identical across experiments and are derived from a randomly selected ensemble member of the FREE case, which is hereafter referred to as TRUTH. To capture the basic influence of observation instrument and algorithmic errors on sea ice DA, observation error magnitudes are expressed as a function of the daily TRUTH value (listed in Table 2). The error magnitude, which can be thought of as the second

moment of a probability distribution, is then used to determine a prescribed observation error distribution (OED) centered on the TRUTH estimate of the observation. Each daily observation is then randomly sampled from the OED. The resulting observation timeseries thus captures reasonable noise around the known TRUTH. In ensemble Kalman DA studies preceding QCEFF, the OED was assumed to be a normal distribution around TRUTH values. Here, the OED is set as a bounded normal distribution, thereby accounting for the physical realities of sea ice observations.

Aggregate observation values extracted are SIT and SIC. The variance of the observation error distribution for each SIT observation is a linear function of the true SIT value on the order of tens of centimeters. Observation error variance for SIC observations is a parabolic function of the true value on the order of ten percent of grid cell area. As a result, observation error magnitudes when SIC declines in the summer months can be quite large, implying a plausible range of observations that may exceed the SIC upper bound of 1. When used to determine a bounded OED that does not exceed 1, these large errors lead to



summer SIC observations that are biased low relative to TRUTH. The ramifications of this bias are discussed in Sections 3 and 3.1.

Categorized observations are also drawn from each of the model's area and volume ITD categories ($A_{\text{ice},n}$ and $V_{\text{ice},n}$) and are always assimilated together (i.e., assimilating $A_{\text{ice},n}$ indicates that each of five area categories are assimilated simultaneously).Categorized area and volume observation error variances are assumed to follow a uniform distribution in each category, weighted by the total area (and midpoint thickness, in the case of volume observations) of that category. These errors are therefore generally less than 10% of the true category value (Fig. 2).

Because sea ice ensembles perturbed only by differing atmospheric conditions (and not by varying model parameters) are generally under-dispersive with respect to SIC (Zhang et al., 2018; Williams et al., 2022; Riedel & Anderson, 2023), we apply enhanced spatially-varying state-space prior inflation (El Gharamti et al., 2019) in each experiment. While the benefits of the spatial variation are lost on our application, the algorithm used implements an inverse gamma function that enables an increase *or decrease* in ensemble spread and outperforms alternative inflation algorithms in some cases (El Gharamti et al., 2019). The applied inflation uses a damping factor of 0.9, a lower standard deviation bound of 0.6, and a maximum per-timestep standard deviation change of 5%.

Spatial localization is practically uninformative in a single-column application, but we explore the effect of "category localization" in the experiments assimilating $A_{\text{ice},n}$ or $V_{\text{ice},n}$. Category localization weights the covariance values between variables in different ITD categories by zero. As a result, an observation from any of individual ITD categories is prevented from updating any state-space variable not also in that same ITD category. In theory, this type of localization should limit the effects of potentially spurious relationships between categories and allow us to more reasonably treat category error variances as uncorrelated.

Finally, since DA is not guaranteed to respect the physical bounds of a system, it is common to use some postprocessing method to correct any non-physical adjustments made by the filter. DART includes three postprocessing options for sea ice: two mass-aware re-scaling approaches and one rebalancing method that has been adapted from a CICE internal function (Riedel & Anderson (2023); the current default in CICE-SCM-DART). All experiments in Table 2 make use of the CICE rebalancing option. Each experiment was rerun using the other two postprocessing methods, but since no significant differences resulted, those additional experiments are not discussed here.

## 2.4 Evaluative Metrics

To evaluate results, the ensemble means of the FREE case and each experiment (EXP) in Table 2 are compared to TRUTH using three metrics: mean absolute error (MAE), root mean square error (RMSE), and the coefficient of efficiency (CE). The presented definitions are generalized such that EXP and TRUTH may represent the experiment ensemble mean and reference "true" value, respectively, of any of CICE-SCM's state or diagnostic variables. In this study, these metrics are applied only to to SIC, SIT, and SND.

MAE measures the average discrepancy between the forecast (FREE or EXP) and TRUTH over the course of the forecast period and is defined as



$$MAE = \sum_{i}^{n} \frac{|EXP_i - TRUTH_i|}{n}, \tag{1}$$

where $n$ indicates the number of timesteps in the forecast period. RMSE, defined as

$$RMSE = \sqrt{\sum_{i}^{n} \frac{(EXP_i - TRUTH_i)^2}{n}}, \tag{2}$$

also evaluates how the forecast deviates from TRUTH but additionally provides a sense of whether the average discrepancy tends to include large outliers. RMSE is therefore always greater than MAE, but in a desirable forecast the difference between the two will be close to zero.

CE (Nash & Sutcliffe, 1970) measures forecast skill compared to TRUTH by evaluating how *efficient* the forecast is as a model of the observed system's mean and variance,

$$CE = 1 - \frac{\sum_{i}^{n} (EXP_i - TRUTH_i)^2}{\sigma_{TRUTH}^2}, \tag{3}$$

and lies between $-\infty$ and 1. CE equal to 1 indicates a perfect match between the forecast and the TRUTH (the numerator in the second term of Eq. 3 is zero), while CE of 0 reflects a forecast that performs only as well as climatological prediction (the deviations of the experiment from TRUTH are equal to the variance of the TRUTH around its mean). A negative CE indicates that the forecast is not skillful. In general, the more positive the CE value, the better the forecast.

To couch results in a generalized framework, differences in MAE and RMSE between the EXP forecasts and the FREE forecast are evaluated using a percent reduction approach, thereby diagnosing the impact of assimilating observations relative to forecast with no assimilation. For example, percent RMSE reduction (pRMSE) due to assimilating observations is calculated as

$$pRMSE = 100 \times \frac{RMSE_{FREE} - RMSE_{EXP}}{RMSE_{FREE}}. \tag{4}$$

CE is a less intuitive metric from an error reduction perspective; the impact of assimilation in this metric is therefore quantified as a CE increase (iCE),

$$iCE = CE_{EXP} - CE_{FREE}. \tag{5}$$

Statistically significant differences between assimilation experiments, FREE, and TRUTH are diagnosed using a Welch's t-test.

## 3 Results

The results of assimilating observations of SIT, SIC, and categorized area $A_{\text{ice},n}$ with an unbounded DA filter (f1_NORM) are presented in Fig. 3. This case illustrates that CICE-SCM-DART replicates the results of larger modeling studies discussed in the





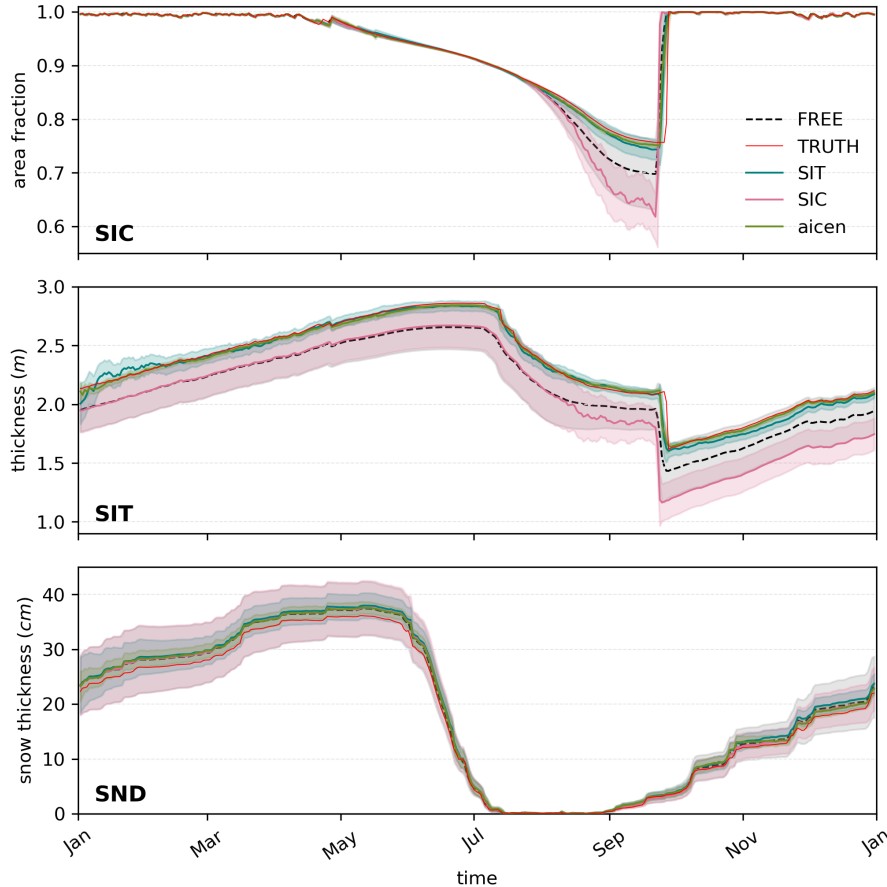

**Figure 3. Assimilating with unbounded algorithms.** The results of using an unbounded filter (f1_NORM) to assimilate SIC (solid pink line), SIT (solid teal) or category area observations (solid green) are shown for modeled SIC (top panel), SIT (middle panel) and SND (bottom panel). The black line represents the FREE case (without assimilation) and the thin red lines are the randomly selected TRUTH. For the results shown, thick lines are ensemble means and shading represents the ensemble standard deviation around the mean. Observations are assimilated at daily intervals throughout atmospheric forcing year 2011.

Introduction. Assimilating SIT observations results in better sea ice analyses year-round than assimilating SIC observations, which have an impact only during the summer months when the model ensemble is capable of capturing variations in sea ice cover. In fact, assimilating SIC observations appears to have a negative impact on modeled SIC in Fig. 3, though this is because our method for producing synthetic SIC observations—which are derived using a bounded normal OED—generates SIC observations that are biased low relative to the TRUTH (Fig. 2). This is particularly true in the summer months when 215 modeled SIC in TRUTH is comparatively low and the prescribed observation error variance is large (Table 2).

Unlike in Fig. 3, when observations are assimilated with a fully bounded filter (f101_BNRH) the bounded observation error distribution is appropriately accounted for and the results lie close to the FREE mean (Fig. 4). We also note that assimilating



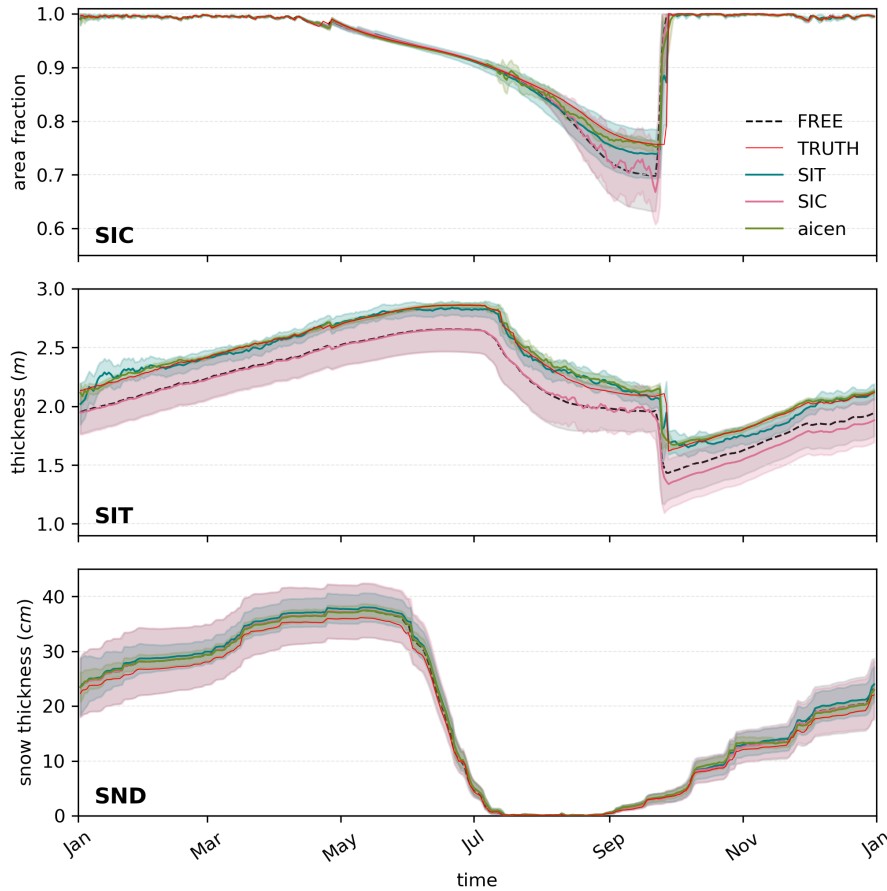

**Figure 4. Assimilating with unbounded algorithms.** Same as Fig. 3 but for case in which observations are assimilated using the f101_BNRH (fully bounded) filter.

SIC observations with $1/10^{th}$ of the error prescribed in Table 2 does shift the resulting modeled SIC closer to TRUTH (not shown), though whether such small magnitude errors are reasonable is a separate discussion left for other work. In contrast,

assimilating $A_{\text{ice},n}$ observations performs at least as well as assimilating SIT observations in the unbounded case, and will be discussed in more depth later.

A more succinct comparison of the experiments listed in Table 2 is presented in Fig. 5. In terms of modeled SIT, we find that the assimilation of any observation that either explicitly or implicitly contains information about ice thickness reduces MAE by between 80 and 95% and improves the CE score by ∼0.3, regardless of the filter used. Assimilating categorized

observations tends to outperform assimilating SIT in terms of pMAE, but falls slightly short according to iCE. This discrepancy across metrics implies that assimilating categorized observations may not capture as much of the observed variance in SIT as assimilating SIT observations, a conclusion supported by the relative pRMSE achieved in each case (Fig. A1). Experiments



assimilating SIT and categorized observations are not significantly different from one another or TRUTH, though they are all significantly different from the FREE ensemble mean (Fig. 6).

Adjustments to modeled SIC are more variable. The relative lack of improvement as a result of assimilating SIC compared to SIT is not a novel result, but a good confirmation that the grid-cell level responses investigated here are reminiscent of sea ice DA studies that use more traditional ensemble filtering methods and assimilate on larger grids. For modeled SIT and SND, there is very little variation in the results as a function of the filter used (Fig. 5). For modeled SIC, larger pMAE tends to stem from cases using unbounded regression methods (NORM) and when assimilating SIC observations to update modeled SIC,
using a bounded filter in observation space (f101) leads to notable improvements compared to using an unbounded filter (f1).

Finally, modeled SND is degraded by the assimilation of sea ice observations in all cases. Assimilating snow depth observations has been shown to improve snow estimates in large models when compared to cases in which snow was updated only via postprocessing (Riedel & Anderson, 2023), as well as in a single-column model when assimilated alongside sea ice observations (Riedel et al., 2023). In the experiments performed here, categorized snow (*vsnon*) is a state variable that is up-
dated via regression with the model's observed quantities but no snow observations are assimilated. The inefficacy of sea ice observations to reduce snow bias likely derives from an ensemble relationship between sea ice variables and categorized snow that produces too much late winter/early spring snow on thicker ice and too little on thinner ice (Fig. 7).

### 3.1    Boundedness

In general, we find the metrics in Figs. 5 and 6 have a rather weak dependence on whether or not the filter respects bounds for
modeled SIT and SND, especially when compared to the obvious dependence on the kind of observation assimilated. There is essentially no dependency highlighted by iCE, and only minimal variation in pMAE. In terms of modeled SIC, however, the impact of using a bounded filter is more apparent (Fig. 8). The use of bounded rank histogram distributions in observation-space allows the filter to correctly infer the bounded nature of the observation error distribution (which respects the physical upper bound of 1 for SIC) and its relationship to TRUTH. The adjustments thus avoid degrading modeled SIC and lead to a
positive annual pMAE (Fig. 5) and reduced bias relative to TRUTH, particularly in the melt season, when SIC observation errors are particularly large (Figs. 8, 4).

The under-performance of bounded filters away from the upper-bound of SIC is likely due to the nature of the model state variables (categorized ice area, ice volume, and snow volume). Recall that the values being diagnosed (SIC, SIT, and SND) are calculated from categorized quantities using forward operators, but are not themselves state variables. This formulation leads
to an issue with properly constraining modeled SIC. In the first step of the assimilation, bounds are placed on the observed quantity, SIC, which is calculated by applying a forward operator (a simple summation) to the model's forecast of the category area fractions in the ITD. Observation space incrementing respects the bounds prescribed on the observable. However, in the second step of the assimilation, the increment calculated between the observation and the model's estimate of the observed quantity is mapped back onto the category-based state variables using regression. This step also respects boundedness, but
must rely on bounds prescribed by the user for each of the state variables. The only objective bounds that can be placed on each individual category area fraction are [0, 1], meaning that the regression of the observation-space increment can update





**Figure 5. Bias reduction and model efficiency as a function of filter type and observation kind.** Percent MAE reduction (pMAE) (left column) and CE increase (iCE) (right column) relative to the FREE forecast as a result of assimilating various observation kinds (x-axis, see Table 2 for definitions) with each filtering method (y-axis). Results are shown for modeled SIC (top row), SIT (middle row), and SND (bottom row). In general, lighter-toned colors indicate a more beneficial impact due to assimilation than darker-toned colors. The number values indicate the specific pMAE or iCE associated with each experiment.



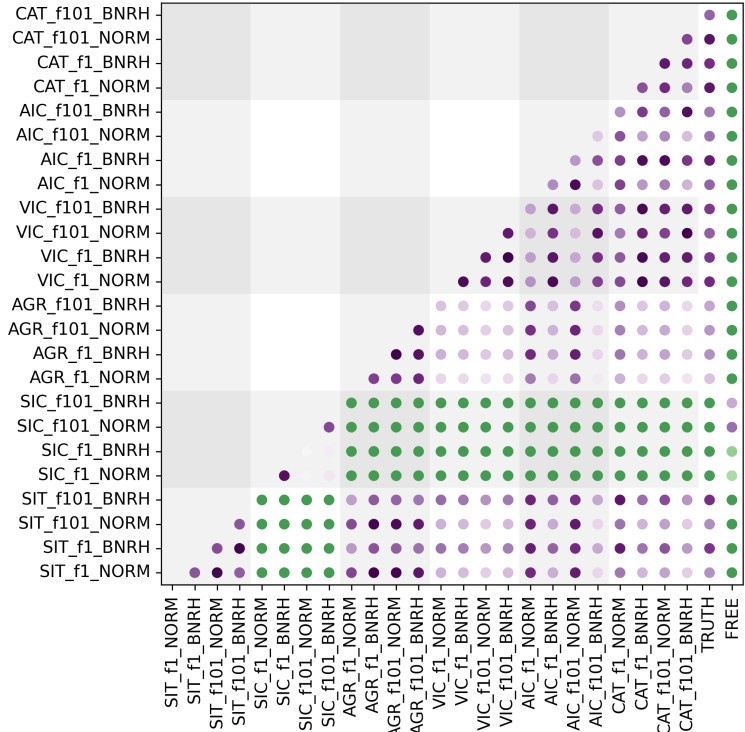

**Figure 6. Significant differences between experiments.** The significance of deviations between each assimilation experiment, TRUTH, and the unassimilated FREE ensemble mean are shown. The color gradient represents p-values for the statistical difference between each experiment shown on the x-axis with each experiment shown on the y-axis. The rightmost columns show the p-values for differences from TRUTH and the FREE case. Purple shades indicate insignificant difference at a p-value of 0.05, while greens indicate that the two cases in question differ significantly at a p-value of 0.05.

each of the individual category area fractions to a value anywhere in that range. However, diagnostic SIC used to evaluate the forecast is calculated anew from the adjusted category area fractions, and is therefore no longer constrained on [0, 1], but rather on [0, 5]. As such, while the bounded filters respect the imposed bounds on both observed and state variables as intended
(not shown),the dependency of the sea ice model on the prescribed ITD categories confounds an attempt to truly respect upper bounds on SIC.

In sum, while the use of bounded assimilation filters does not produce significantly better or worse results in terms of the impact on modeled SIT or SND, some improvements are carried through for modeled SIC. While the full impact of boundedness in filtering is limited in this study, these filters could still provide a path to eliminating postprocessing if further
infrastructure designed to simultaneously constrain SIC and categorized area in CICE-SCM-DART were developed.



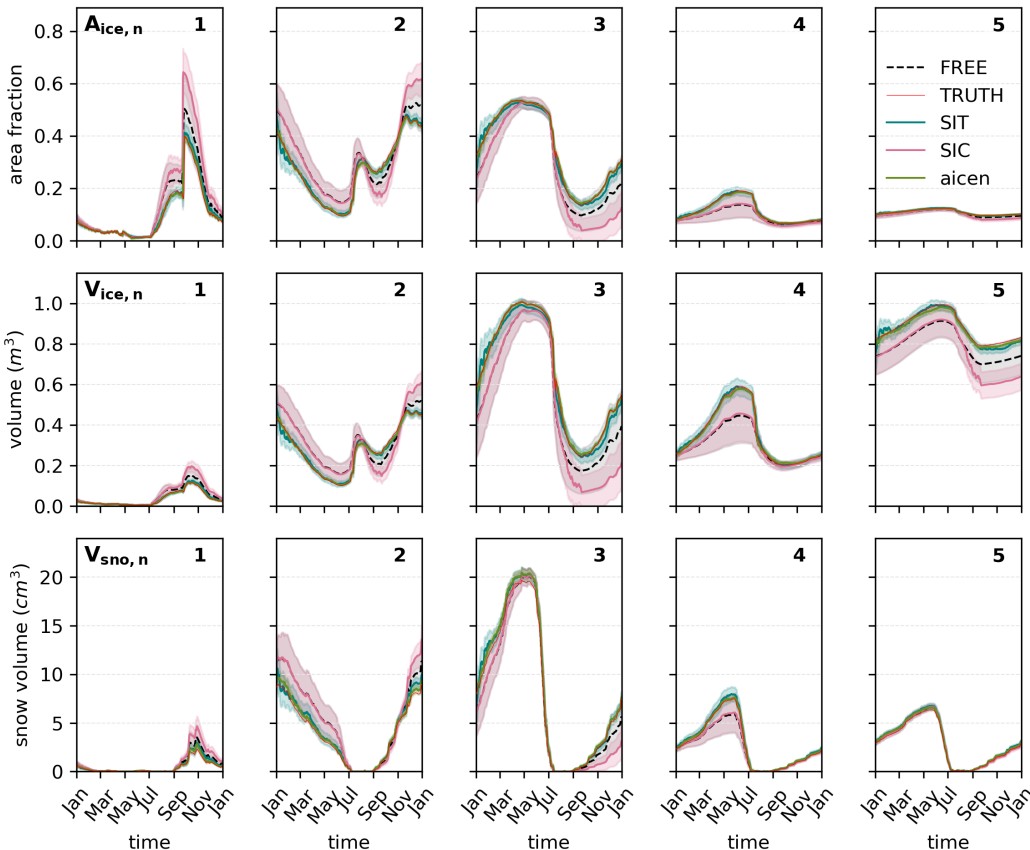

**Figure 7. Category-level impact of assimilating with unbounded algorithms.** Same as Fig. 3 but for each of the model's area (top row), volume (middle row) and snow volume categories (bottom row). TRUTH (the thin red line) may be difficult to identify in some panels, as the cases assimilating SIT and $A_{ice,n}$ (solid teal and green lines) lie very close to TRUTH.

## 3.2 Category Assimilation

More so than constraining the data assimilation with bounded filters, assimilating the model's categorized ice thickness distribution directly improves the results. First, assimilating categorized area or volume (or both) tends to lead to higher MAE reductions in modeled SIT and SIC, particularly in the cases that used unbounded regression in the multivariate adjustments

(Fig. 5a,c). Additionally, while modeled SND is found to be degraded in all cases presented here, categorized observations significantly reduce the degree to which assimilating ice observations increases the MAE (Fig. 5e).

There also appears to be evidence that assimilating categorized observations may consistently constrain the sea ice state regardless of the mean state grid cell thickness. In Fig. 7, assimilating SIT observations and categorized area observations perform comparably to constrain a categorized sea ice state that is relatively thick and thus has a non-negligible amount of

ice in each category, including the thickest. By comparison, Fig. 9 (bottom row) presents a comparison case in which we





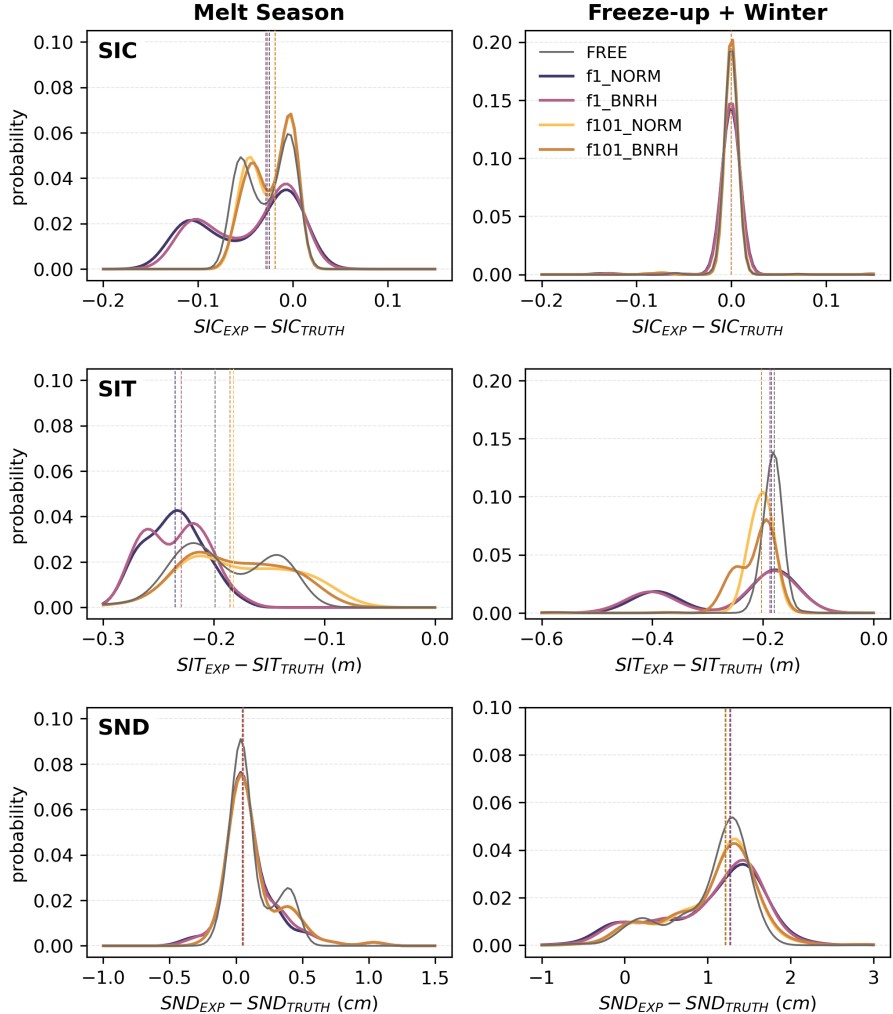

**Figure 8. Seasonal impact of using bounded filters when assimilating SIC observations.** The normalized probability density functions of the differences between the EXP mean and TRUTH in each assimilation cycle are shown for modeled SIC (top row), SIT (middle row) and SND (bottom row), along with their respective sample medians (dashed vertical lines). The dark grey distribution in each panel represents the difference between the FREE ensemble mean and TRUTH as a reference. The differences are divided into the melt season (left column, July 1st - September 15th) and the rest of the year (right column) and highlight the positive seasonal impact of using bounded algorithms—in the melt season, the distributions are shifted closer to a EXP-TRUTH difference of zero when a bounded filter is used (yellow lines) than when an unbounded one is used (pink lines). This effect is most prominent in the melt season months because the uncertainties associated with the assimilated SIC observations are largest in these months, and thus the bounded synthetic observations are more biased relative to TRUTH. The bounded algorithms correct for this appropriately.



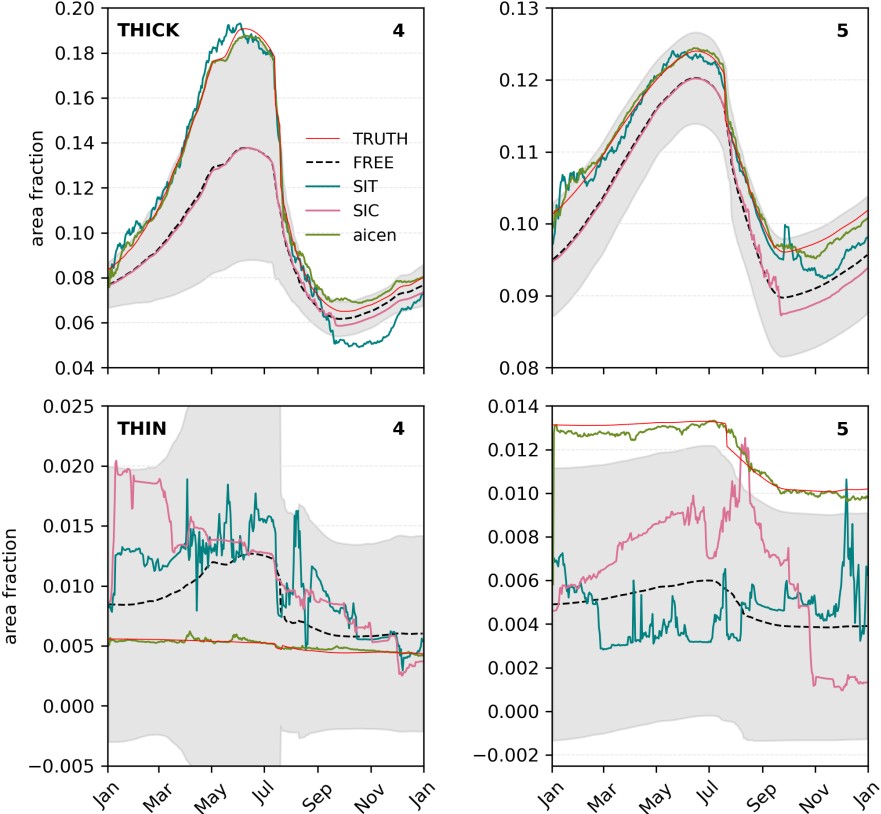

**Figure 9. Category-level impact of assimilating in a thin vs thick ice state.** The model's two thickest area categories for the standard case (top row, THICK) are repeated from Fig. 7. A corresponding experiment in which mechanical ridging is restricted (bottom row, THIN), leading to very low concentrations of ice in the thickest ITD categories is also shown. These results demonstrate that the strength of the relationship between SIT observations and modeled $A_{\text{ice},n}$ breaks down when some categories have very little ice and that this can bias the modeled SIC result.

have restricted CICE-SCM's ability to mechanically ridge ice, thereby preventing the buildup of ice in the thickest two ice categories. This leads to an overall thinner mean state, qualitatively similar to a first-year ice regime. In this thinner ice case, SIT observations cease to constrain the thick ice categories (4 and 5). While the erroneous adjustments made in the thickest two ice categories are relatively minimal compared to the total grid-cell mean SIT, we do find that they lead to noticeable low

biases in modeled SIC (not shown). Assimilating categorized area observations appears to avoid this issue entirely (Fig. 9, top row)—the modeled quantities produced by doing so are consistent with TRUTH in all categories and total SIC.

    There exist at least two potential applications of this finding for more realistic experiments. First, in more practical applications, the assimilation of categorized variables may avoid introducing small errors in low-concentration ITD categories that occur when assimilating SIT and thus mitigate the overall error propagation of the model during intervals in which real-world





SIT observations are historically unavailable (i.e. during summer months). Second, it has been noted in previous work that assimilating SIT can lead to biases in the sea ice edge (Riedel & Anderson, 2023), which introduces an incentive to assimilate SIC as well as SIT, despite the negative impact SIC can have on modeled quantities away from the ice edge. The consistency resulting from assimilating categorized observations in multiple ice states, including thin, first-year type regimes, suggests a better solution for constraining the sea ice state everywhere in the Arctic.

## 4  Discussion

This study confirms that assimilating SIT observations improves sea ice analyses over assimilating SIC observations at the grid cell level. In these experiments, assimilating SIT followed by SIC leads to comparable but slightly degraded modeled quantities when compared to just assimilating SIC, which implies that for this ensemble and mean state, there is very little benefit to assimilating SIC observations, especially outside the boreal summer season. This finding, which applies to SIC
analyses as well as SIT, may be due in part to the fact that we have generated spread in our ensemble using only variable atmospheric forcing and that the ensemble is under-dispersive with respect to SIC for much of the year. It is worth noting, however, that in this under-dispersive SIC scenario, assimilating SIT still improves modeled SIC.

An emergent finding of this work is the positive impact of assimilating the categorized state (the ITD). Assimilating categorized area and volume estimates reduces MAE and increases CE on par with assimilating SIT observations, improving the
model's estimates of SIC and SIT at the category level even when the mean ice state is thin and some categories contain very little ice. Assimilating categorized observation also reduces the forecast error beyond that of assimilating aggregate observations (Figs. 3, 7), though this is likely related at least to the fact that the categorized observation errors can be quite small (see *Methods*).

The application of a series of bounded filtering algorithms is novel to sea ice data assimilation and has highlighted the
complexities of assimilating observations into a categorized distribution model such as CICE-SCM. The sometimes negligible impact of bounded algorithms on modeled sea ice quantities indicates a need to further tailor the CICE-SCM-DART interface such that the filters constrain categorized variables and SIC simultaneously. Bounded algorithms eliminate any need for postprocessing of $V_{\mathrm{ice},n}$, SIT, or $V_{\mathrm{sno},n}$ (not shown). However, for modeled SIC we find that bounded algorithms result in a small fraction of the adjustments made requiring SIC postprocessing (Fig. 10). Note that assimilating categorized observa-
tions reduces postprocessing requirements compared to assimilating aggregate observations, likely because the categorized observations are closer in nature to the model's state variables.

While the broad strokes of these results are expected to carry over to assimilating real-world observations, the details are likely to vary under replication in larger models, where dynamic exchange between grid cells imbues additional information into the observation-state relationships and introduces the need for localization in the data assimilation framework. We also
acknowledge that the bounded filtering algorithms employed in this work depend on piece-wise distributions that are a function of the model ensemble and are relatively uninformed otherwise. DART provides the opportunity to use alternative distributions that may qualitatively shift the results. Finally, the work presented here avoids the role of various forms of model error that





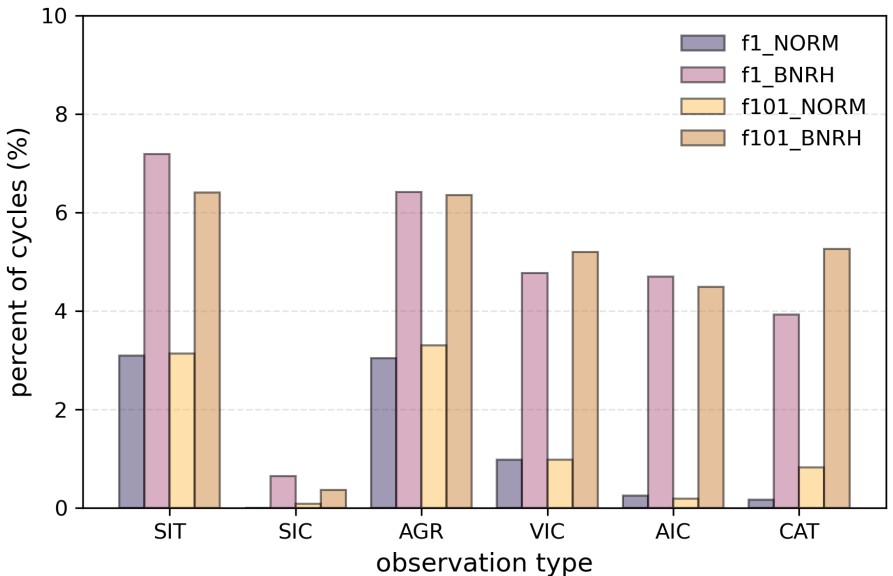

**Figure 10. Percent of DA cycles requiring SIC postprocessing.** For each type of observation assimilated with the four filters (f1_NORM, f1_BNRH, f101_NORM, or f101_BNRH), the percent of total DA cycles over the course of each experiment that resulted in non-physical values of aggregate SIC are shown. Note that the percent of postprocessing when assimilating SIC is artifically small, as those observations do not lead to substantial adjustments from the FREE mean except in the summertime, when grid cell aggregate SIC is decidedly lower than the upper bound (no postprocessing occurs during this time).

are present in operational data assimilation, where the observations and evolution between them are unlikely to be correctly captured by forward operators and model physics. Therefore, at the very least, the magnitude of error reductions in sea ice
analyses presented may overestimate what will be achievable in more practical applications.

## 5   Conclusions

We have interrogated in detail the grid-cell level response of a complex sea ice model to the assimilation of various kinds of sea ice observations, including SIT, SIC, and categorized area and volume, and found that SIT and categorized observations most accurately constrain the ensemble mean forecast in both category ITD state variables and diagnostic grid-cell mean
SIT and SIC; categorized observations are the only observations that perform consistently well across two different grid cell mean thickness states. Two key issues in the application of bounded data assimilation algorithms to the sea ice problem are identified. First, an approach to appropriately constrain categorized area and total SIC simultaneously is still needed. Secondly, a true understanding of where and *why* assimilation improves (or degrades) model estimates of the sea ice state depends on how well the model ensemble captures natural covariance relationships between observables and state variables, a targeted study of
which is absent from previous literature due to a lack of necessary observations. Future work will attempt to address the first



issue and diagnose the second. Assuming that the ensemble is reasonably realistic in terms of relationship between variables, the findings presented here are expected to be qualitatively consistent in larger grid models and more practical assimilation experiments.

*Code and data availability.* All code used in the study can be found on Github. The CICE5 single column model is available from the CICE
Consortium at https://github.com/CICE-Consortium/CICE. The Data Assmilation Research Testbed is maintained by DAReS and hosted at https://github.com/NCAR/DART. The version of DART used for this study was forked to https://github.com/mollymwieringa/DART. The python scripts and Jupyter notebooks used to configure, run, and evaluate the experiments in this study have been collected in a separate Github repository (https://github.com/mollymwieringa/cice-scm-da); the postprocessed experiment data used to produce the figures is available upon request.

*Author contributions.* All authors contributed to the conceptualization of the study. MMW performed the experiments, analyzed the results,
and wrote the manuscript. MMW, CMB, and CPR each contributed to developing the CICE-SCM-DART interface. JLA led the development of the bounded data assimilation algorithms and consulted on their application in this study.

*Competing interests.* The authors declare no competing interests.

*Acknowledgements.* This material is based upon work supported by the National Center for Atmospheric Research, which is a major facility
sponsored by the National Science Foundation under Cooperative Agreement No. 1755088. CMB and MMW acknowledge support from NASA ROSES Grant Number 80NSSC21K0745. MMW was also supported in part by the University of Washington College of the Environment's Integral Environmental Big Data Research Fund. The authors would like to specially thank the DAReS team at NCAR/CISL, Ian Grooms, Alek Petty, Jon Poterjoy, and David Bailey for many insightful conversations and helpful technical support. We thank the editor and anonymous reviewers for constructive comments that helped improve the manuscript.





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



**Table 1.** QCEFF filter components. Includes the distributions used to represent the model ensemble in observation-space incrementing and state-space regression; the observation error distribution associated with each filter; any DART filter equivalents; and relevant references.

| Filter Name | Obs. Space Dist. | State Space Dist. | Obs. Error Dist. | DART Filter Equivalent | References |
|---|---|---|---|---|---|
| f1_NORM | normal | normal | bounded normal | EAKF | Anderson (2001) |
| f1_BNRH | normal | BNRH | bounded normal | *none* | *none* |
| f101_NORM | BNRH | normal | bounded normal | BRHF | Riedel et al. (2023) |
| f101_BNRH | BNRH | BNRH | bounded normal | *none* | Anderson (2023) |



**Table 2.** Assimilation experiments. Filter refers to the filter type used (see Table 1); Obs. Kind to the type of observation assimilated; and Obs. Error to the formula used to determine an individual error estimate for each observation at each timestep. $a_n$ and $v_n$ refer to the area and volume in the $n^{th}$ ITD category; $\overline{h_n}$ refers to the midpoint thickness in the same category. $h_R$ and $h_L$ are the upper- and lower-most thickness bounds used to define the $n^{th}$ ITD category.

| Case Name | Filter | Obs. Kind | Obs. Error |
|---|---|---|---|
| SIT_f1_NORM | f1_NORM | SIT | $\sigma_{SIT} = 0.1 SIT$ |
| SIC_f1_NORM | f1_NORM | SIC | $\sigma_{SIC} = -0.5(SIC^2 - SIC)$ |
| AGR_f1_NORM | f1_NORM | SIT, SIC | $\sigma_{SIT}, \sigma_{SIC}$ |
| AIC_f1_NORM | f1_NORM | *aicen* | $\sigma_{a_n} = (\frac{a_n}{\overline{h_n}})^2 \frac{(h_R - h_L)^2}{12}$ |
| VIC_f1_NORM | f1_NORM | *vicen* | $\sigma_{v_n} = (a_n)^2 \frac{(h_R - h_L)^2}{12}$ |
| CAT_f1_NORM | f1_NORM | *aicen, vicen* | $\sigma_{a_n}, \sigma_{v_n}$ |
| SIT_f1_BNRH | f1_BNRH | SIT | $\sigma_{SIT} = 0.1 SIT$ |
| SIC_f1_BRNH | f1_BNRH | SIC | $\sigma_{SIC} = -0.5(SIC^2 - SIC)$ |
| AGR_f1_BNRH | f1_BNRH | SIT, SIC | $\sigma_{SIT}, \sigma_{SIC}$ |
| AIC_f1_BNRH | f1_BNRH | *aicen* | $\sigma_{a_n} = (\frac{a_n}{\overline{h_n}})^2 \frac{(h_R - h_L)^2}{12}$ |
| VIC_f1_BNRH | f1_BNRH | *vicen* | $\sigma_{v_n} = (a_n)^2 \frac{(h_R - h_L)^2}{12}$ |
| CAT_f1_BNRH | f1_BNRH | *aicen, vicen* | $\sigma_{a_n}, \sigma_{v_n}$ |
| SIT_f101_NORM | f101_NORM | SIT | $\sigma_{SIT} = 0.1 SIT$ |
| SIC_f101_NORM | f101_NORM | SIC | $\sigma_{SIC} = -0.5(SIC^2 - SIC)$ |
| AGR_f101_NORM | f101_NORM | SIT, SIC | $\sigma_{SIT}, \sigma_{SIC}$ |
| AIC_f101_NORM | f101_NORM | *aicen* | $\sigma_{a_n} = (\frac{a_n}{\overline{h_n}})^2 \frac{(h_R - h_L)^2}{12}$ |
| VIC_f101_NORM | f101_NORM | *vicen* | $\sigma_{v_n} = (a_n)^2 \frac{(h_R - h_L)^2}{12}$ |
| CAT_f101_NORM | f101_NORM | *aicen, vicen* | $\sigma_{a_n}, \sigma_{v_n}$ |
| SIT_f101_BNRH | f101_BNRH | SIT | $\sigma_{SIT} = 0.1 SIT$ |
| SIC_f101_BNRH | f101_BNRH | SIC | $\sigma_{SIC} = -0.5(SIC^2 - SIC)$ |
| AGR_f101_BNRH | f101_BNRH | SIT, SIC | $\sigma_{SIT}, \sigma_{SIC}$ |
| AIC_f101_BRNH | f101_BNRH | *aicen* | $\sigma_{a_n} = (\frac{a_n}{\overline{h_n}})^2 \frac{(h_R - h_L)^2}{12}$ |
| VIC_f101_BNRH | f101_BNRH | *vicen* | $\sigma_{v_n} = (a_n)^2 \frac{(h_R - h_L)^2}{12}$ |
| CAT_f101_BNRH | f101_BNRH | *aicen, vicen* | $\sigma_{a_n}, \sigma_{v_n}$ |



## Appendix A

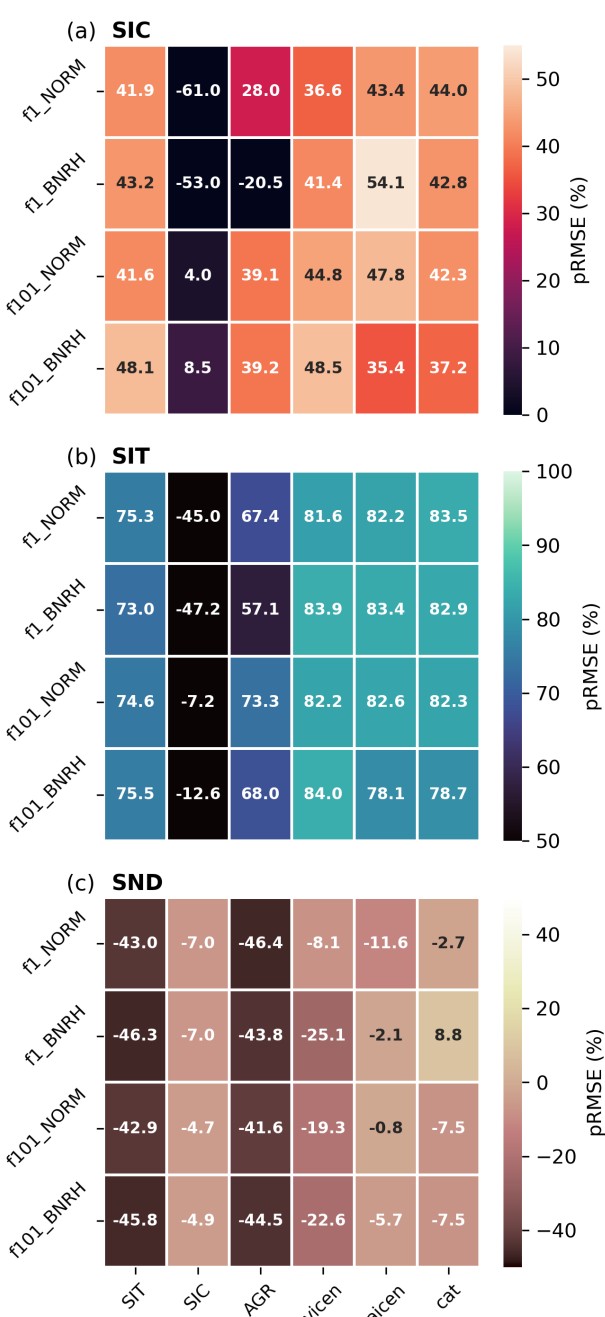

**Figure A1. RMSE reduction as a function of filter type and observation kind.** Same as Fig. 5 but for percent RMSE reduction (pRMSE).