# Peer review of "Bounded and categorized: targeting data assimilation for sea ice fractional coverage and non-negative quantities in a single column multi-category sea ice model"

_EGUsphere, 2023_

## Author Comment (AC1)

**Bounded & categorized: targeting data assimilation for sea ice fractional coverage and non-negative quantities in a single column multi-category sea ice model**

*Manuscript egusphere-2023-2016*

**Responses to Referee 1**

The authors would like to thank the editor and Referee 1 for the time and effort that have gone toward providing feedback on this manuscript. Please find below, in blue, our responses to referee comments, questions, and concerns. All page numbers refer to the revised manuscript.

*General Comments*

Bounded and categorized: targeting data assimilation for sea ice fractional coverage and non-negative quantities in a single column multi-category sea ice model" tests two data assimilation approaches on single column sea ice model (Icepack, the column physics of CICE) in a perfect-model framework. Assimilating per-category quantities (ice area and volume) is found to substantially improve the skill of the modeling system to be on par with assimilating sea ice thickness and superior to assimilating concentration: which, in the standard, unbounded approach actually performs worse than conducting no assimilation. Bounded data assimilation only shows benefits compared to unbounded when assimilating concentration, and even then, the results are no better than conducting no assimilation. For other parameters bounded assimilation does not yield a net improvement, because the 'unbounded' cases are effectively bounded in post-processing. The results, especially the per-category ice area findings, are of broad interest to the sea ice community and merit publication in The Cryosphere.

However, there are scientific, technical, and communication issues that need to be addressed before publication.

1. Scientific: the primary scientific issue that needs to be addressed is that the "observational uncertainties" applied to ice concentration, thickness, and category variables need to be scientifically justified for the findings to be meaningful. It would maximize the impacts of the work if the justification were based on our real or anticipated observational capabilities (e.g., what are the uncertainties in satellite-derived sea ice concentration). However, if this is not feasible, the rationale for the prescribed uncertainties needs to be clearly explained in the text.

2. Scientific: a minor point, the Discussion suggests that the findings are valid in a seasonal (first-year) ice regime, but no simulations from a first-year ice regime are presented. I recommend limiting the Discussion to what is clearly supported by the presented results (i.e., not making

claims about FYI). Alternatively, if the authors choose to increase the scope of the work to include FYI simulations (which will require a different atmospheric and oceanic forcing) that would also be acceptable.

3. Technical: the authors do not cite which version of the Icepack code they used, but the plots suggest that the code contains a known error in the mechanical forcing that substantially changes the simulated ice concentration (https://github.com/CICE-Consortium/Icepack/pull/433). If so, the simulations need to be repeated using a version of the model that doesn't contain this error.

4. Communication: it would enhance the impact of this work if the description of the methods were accessible to a general reader of The Cryosphere who is not an expert in data assimilation. As written, I struggled to follow the critical details of the data assimilation procedure. I have tried to provide guidance here in the detailed comments below, but I recommend having a colleague who is not an expert at DA take a close read of the manuscript before resubmitting.

**Author response:** We thank Referee 1 for the summary of their concerns presented at the outset of their comments and for their conclusion that the results presented in our manuscript merit publication. Our point-by-point responses are included below.

Detailed comments:

Line 25: please provide a reference for the climate model application of DA.

**Author response:** Reference to Brennan & Hakim (2022), who assimilated surface temperature proxy observations into an ensemble of CMIP5 Last Millennium model simulations to reconstruct annual sea ice concentration in the Common Era, has been added to Line 25.

Lines 29-34: The explanation of the steps of DA is hard to follow for someone who does not regularly use DA. For example, it's not clear how the second step "updating the model's estimates of the observed quantity" relates to the first "generate an ensemble of forecasts". Does "update" refer to changing the state variables in each of the ensemble members, selecting the ensemble member that most closely matches the observations, or something else? Because there are numerous Kalman-related data assimilation techniques, it would be helpful if the explanation here could be geared towards a reader without a background in DA and include a representative example of applying the steps in this work (e.g., updating per-category ice concentration, or similar).

**Author response:** We thank the referee for highlighting the need for more effective communication regarding the data assimilation methods. To address the specific question regarding the use of the term "update", in this context "updating the model's estimate of the observed quantity" refers to changing the model's diagnostic estimate of the observation (e.g. SIC calculated from the model's categorized area fraction state variable, $A_{ice,n}$). The update to the model's estimate of the observation (SIC) is then used to determine how much the state variable ($A_{ice,n}$) needs to be updated, but these are separate steps in the process (observation-space incrementing versus state-space regression). We have added a schematic figure (Figure 1 in the revised manuscript) that outlines a data assimilation cycle that is agnostic to the specific Kalman filter type and attempts to frame the cycle in terms of sea ice quantities. We have also updated the initial explanation of the data assimilation cycle in Lines 29-37 to be consistent with the new figure. Additional detail is also available in the Figure 1 caption.

Line 44: Is the term "analysis inaccuracy" specific jargon in DA? Define if so. Otherwise, I think the term "bias" is more readily understood (by the modeling community at least) to mean a systematic error in an ensemble mean.

**Author response:** "Analysis inaccuracy" is not a common term in DA though "analysis" is a specific term that refers to the result produced by the DA filter. "Bias" is involved in many different aspects of the DA process (observation bias, forward operator bias, model bias...). However, we agree that "bias" is a more readily understood term than "analysis inaccuracy." As such, we have replaced "*analysis inaccuracies*" with "*a bias in the assimilation analysis*" in Line 47.

Line 47: The use of "Secondly..." confused me because I don't see a "firstly". I'd recommend a different transition.

**Author response:** Thanks for pointing this out! The use of "*First*" in Line 38 of the original manuscript has been updated to "*Firstly*" in Line 41 of the revised manuscript.

Line 50: Mechanical properties as well as thermodynamic.

**Author response:** Thanks for catching this oversight! As suggested, we have included reference to "*thermodynamic and mechanical processes that are strongly dependent on ice thickness*" in Line 53 of the revised manuscript.

Lines 57-61: Related to the comment on Lines 29-34, a flow-chart or some other figure displaying the steps of DA as used in this work, and how aggregation and boundedness play a role in those steps would

make this clearer for readers. This flow-chart could also highlight the innovation of this work compared to prior research.

**Author response:** As suggested by the referee, and in response to the related comments, we have added such a schematic figure to the manuscript (Figure 1). The figure is more explicit about the conversion of model state variables to observed quantities ("aggregation") and about how sea ice bounds might lead to non-physical updates during the filtering processes.

Line 81: Using the term "observed" for a quantity produced entirely synthetically seems likely to cause confusion with readers. Is there another term in the DA literature that can be used? If not, it seems worth defining one.

**Author response:** Within the DA literature, both synthetic and real observations are described as such, as the intent of studies using synthetic observations is often to gain some understanding of the impact of assimilating measurements from current or potential future observing systems. To avoid introducing jargon unfamiliar to either data assimilation or sea ice research communities, we opt to proceed with our use of "observed" and "observation" when referring to the synthetic observations used in this study. However, we clarify in both the Abstract and in Section 2.3 that the DA experiments being performed are Observing System Simulation Experiments (Line 6 and Lines 143-147) and stress the nature of the observations as synthetic in Section 2.3 (Lines 150-170).

Lines 84-86: The description of the different grid cells is slightly misleading unless this is using a non-standard version of Icepack. The "open ocean", "slab ice", and "categorized ITD" cells all represent the ice cover using an ITD. The difference is purely in the initial conditions (e.g., for slab all, of the ice starts in a single thickness category). The different grid cells are irrelevant to this work so I would just remove the description.

**Author response:** Per the referee suggestion, we have removed the description of Icepack's additional grid cells, as they are indeed irrelevant to this study.

Lines 88-89: What does "tuned to the atmospheric forcing" mean? I generally think of tuning as bringing the model state in line with observations, but that is not apparent in this description.

**Author response:** Please see the response to the next referee comment, which is of a related nature.

Line 90: The phrase "discontinuous behavior in ice concentration related to ice-albedo feedback during the melt season" needs to be explained. How large of a change in ice concentration in a time step is required to be "discontinuous"? If this is a bug in the model it needs to be fixed, as it may introduce other more subtle errors that are not remedied by setting R_snw = -2. If it is expected behavior for the model, why exclude it? Note that Figure 1 shows what appears to be discontinuous behavior during freeze-up.

**Author response:** By 'discontinuous behavior,' we refer to a tendency in our Icepack configuration toward a rapid refreezing event after ice concentration has begun to decline in early June, based on the prescribed value of $R\_snw$. Following the rapid refreezing, and unlike the expected rapid refreezing in September, ice concentrations stay fixed at 100% ice coverage *for approximately two weeks before dropping equally rapidly* to the value prior to the original refreezing. We have included a figure that illustrates this behavior for a range of prescribed $R\_snw$ values with a single atmospheric forcing. We found that for values of $R\_snw$ > -0.5 (blue members), simulations experienced this behavior, but for values of $R\_snw$ < -0.5 (red members), simulations experienced no such rapid refreezing.

The role of $R\_snw$ in the sea ice model is to adjust the size of snow grains—more negative values of $R\_snw$ result in larger snow grains and a lower snow albedo, while more positive $R\_snw$ leads to smaller grains and more reflective snow. When $R\_snw$ is set to -2 (a fairly large snow grain size and low snow albedo) and each simulation is forced with different atmospheric forcing, we produced an ensemble that does not exhibit this undesirable behavior, and which has members nicely distributed across a wide range of summertime sea ice concentrations. (Figure 2 in the revised manuscript). We believe that the undesirable behavior may a kind of shock ice-albedo effect that arises during the melt season due to a handoff between the albedo of melting snow and the underlying sea ice. It is also possible that our chosen combination of atmospheric forcing, prescribed ocean initial conditions, and

dynamics forcing set the stage for such an effect, hence our use of the phrase "tuned to the atmospheric forcing". For the purposes of this work (and because we've selected an *R_snw* value that has been used in configurations of CICE5 coupled to an atmospheric model), the choice of *R_snw* is not particularly relevant to the results, beyond its role in producing a reasonable sea ice ensemble. To avoid unnecessary confusion, we have recast discussion of the *R_snw* parameter in the revised manuscript. The text

> *"The sea ice model is tuned to the atmospheric forcing by setting the snow grain radius parameter (R_snw) to a value of −2. This choice prevents discontinuous behavior in ice concentration related to ice-albedo feedback during the melt season and thus allows for a wider summertime ensemble spread. The number of categories used in the ITD is set to 5."*

has been replaced with

> *"The number of categories used in the ITD is set to 5. The snow grain radius parameter (R_snw) is set to a value of −2. This choice, which is among the default values of R_snw used when CICE is coupled to an atmospheric model, avoids rapid refreezing events during the melt season that lead to unreasonably high summertime sea ice concentrations given the atmospheric forcing conditions."*

in Lines 97-101.

Line 92: The default values in Icepack and the code itself change over time. Please cite the specific tag or release of the code used. Also, it would help the casual reader to specify a few more details about the Icepack setup: at minimum which schemes are being used for thermodynamics, shortwave, and melt ponds. Is a dynamics forcing being used? If so, it should be cited. Relatedly, I suspect the version of Icepack that was used in this work contained an error related to how the dynamics forcing was implemented (https://github.com/CICE-Consortium/Icepack/pull/433). Note how in the summertime in the Fig. 2 TRUTH simulation (a single ensemble member) the ice concentration and thickness change monotonically and there is no evidence of synoptic scale variations due to ice divergence and convergence (which are present in the SHEBA ice dynamics forcing data that are often used to test Icepack). This error substantially changes the simulated ice concentration. If so, the results need to be recomputed without the error.

**Author response:** Thank you for the comment regarding the specifics of the model version! We have included a reference to the version number (1.3.1) and appropriate citation in Line 81. Additional details regarding the model configuration have been added per the referee's requests to the first paragraph of Section 2.1, Lines 92-98, which now read,

*"Icepack is maintained as the column physics module of CICE, with consistent thermodynamics, mechanical redistribution, and tracer support. For use in the CICE-SCM-DART framework, 30 instances of Icepack are forced by unique atmospheric conditions extracted from randomly selected members of a recent large-ensemble reanalysis product (Raeder et al., 2021). Each instance of Icepack uses the mushy thermodynamics scheme (kitd = 1) and linear ITD remapping options (ktherm = 2), as well a Delta-Eddington shortwave radiative transfer scheme and the empirical CESM melt pond scheme. Dynamical forcing to the column is provided by sea ice deformation rates obtained from the SHEBA field campaign (Lindsay , 2002). The number of categories used in the ITD is set to 5."*

To address the portion of the comment noting a potential bug related to the implementation of the dynamics forcing, we integrated the code changes within the highlighted pull request (https://github.com/CICE-Consortium/Icepack/pull/433) into our Icepack setup (version 1.3.1) and produced two alternative ensembles. The first fluxes in ice of identical distribution when ice converges and leads to a drop in grid cell ice concentration (*fluxice*); the second fluxes in open water in the same scenario (*fluxwat*).

First, we note that neither exhibits substantially more synoptic scale variability than our original ensemble. We hypothesize that this is because we are running Icepack in a somewhat atypical manner, with our own calculated ocean initial conditions for temperature, salinity, and heat flux *as well as* a prescribed dynamics forcing from the SHEBA campaign. The model is typically run with one or the other. However, we don't expect that more synoptic scale variations in our ensemble members would significantly alter our assimilation results.

Secondly, we find that **including the suggested code changes did alter the mean state into which we assimilated observations but did not qualitatively change our results**. The *fluxice* ensemble runs away to dramatic ice thicknesses (~60m) during the 10-year spinup period and was deemed ineligible for assimilation. The *fluxwat* ensemble is more stable and better captures the nature of the location as a seasonal ice zone (ice concentrations in several members fall all the way to zero in the summer months). As such, we reran our full suite of assimilation experiments in the *fluxwat* ensemble and have updated all the figures in the revised manuscript with the results. Our conclusions regarding the relative impact of each assimilated observation kind, as well as of the efficacy of bounded versus unbounded assimilation algorithms, remain the same.

Lines 92-93: How is mixed layer salinity being prescribed? The mushy layer physics can be sensitive to this.

**Author response:** If by 'mixed layer salinity' the referee refers to the salinity of the ocean below the sea ice as it pertains the formation of new ice or ice freezing/melt processes, (sea surface salinity, 'sss'), then salinity is prescribed in the ocean initial conditions, which were calculated from the ocean component output of a fully-coupled CESM2 historical simulation. If the referee refers to something else, we will need additional clarification to address this question.

Figure 1: the right middle and bottom plots appear to be incorrect. If the y-axes are the same as on the left plots, then the thickness (ice or snow) in each category is less than the mean, which cannot be. It looks like what is plotted is actually per-category volume. I recommend plotting the per-category thickness as that is more intuitive for readers who are not CICE experts and more directly comparable with observations. Also, the ice thickness (and maybe snow depth?) appear to be ice-area weighted quantities (I'm guessing from the sudden drop during freeze-up). If so, this needs be stated explicitly.

**Author response:** We thank the referee for noting the inconsistency. The right middle and lower plots in Figure 2 of the revised manuscript show per-category ice and snow volume, respectively. The choice to present per-category volumes, instead of thicknesses, is intentional, because per-category ice area, ice volume, and snow volume are the state variables in the model that are updated during Step 4 of the DA cycle (state-space regression). Relatedly, we plot area-weighted grid-cell mean ice and snow thicknesses in the left middle and bottom plots because these are the diagnostic model estimates of the observations that will be assimilated (quantities that might be updated in Step 3 of the cycle, observation-space incrementing). Y-axis labels for the left column of plots have been added, to clarify that they are category volumes, not thicknesses. The figure title now reads "*The FREE ensemble's three aggregate variables and the state variables from which they are derived*" and the figure caption includes an explicit mention of the area-weighted nature of the aggregate variables (Figure 2).

Line 95: Why this particular location? If the sea ice is supposed to seasonally advance and retreat from this location, then why doesn't Figure 1 show ice thickness and area fraction declining to zero (or at least very close) in the summer?

**Author response:** The use of a seasonal location for this case study allows use to examine how well the bounded DA methods perform near both the upper and lower bounds of sea ice concentration (0 and 1), as well as the lower bound sea ice thickness (0). The referee's suggestion to ensure that the dynamics forcing is implemented correctly in our model setup results in an ensemble that better reflects the seasonal nature of the selected location (Figures 2 and 3 in the revised manuscript).

Line 97: Why conduct a 10 year spin up? Is it needed for the DA?

**Author response:** The spin-up period allows the sea ice conditions across the ensemble to diverge due to the difference in atmospheric forcings. Ensemble spread, which can be interpreted in this case as a sampling of natural variability in sea ice conditions due to atmospheric variability, is crucial for the data assimilation algorithms—if there is too little spread in the ensemble, then the algorithm will produce an analysis that leans too heavily on the model estimate of the state and insufficiently accounts for the observations. If there is no spread in the ensemble, the DA filter cannot adjust the model using observational information at all. We have added a phrase ("... allowing the sea ice simulations to diverge in response to atmospheric variability") to this sentence in Lines 107-108 of the revised manuscript.

Line 105: See comments above related to making the steps of DA more understandable for the non-expert.

**Author response:** Per this comment, and the earlier comments related to explanation of the DA, Section 2.2 has been updated to reference and be consistent with the newly included Figure 1 (a schematic outline of the data assimilation cycling process) (Lines 116-117).

Line 109: Increases relative to what?

**Author response:** The EAKF (Anderson 2001) enables stable and efficient assimilation with smaller ensembles sizes when compared to the traditional Ensemble Kalman Filter (Evensen, 2003). To clarify, *"... increases the stability and efficiency of assimilating with smaller ensemble sizes"* has been replaced with *"...increases the stability and efficiency of assimilating with smaller ensemble sizes compared to a traditional EnKF"* in Lines 120-121 to clarify.

Lines 137-142: What is the basis for these observational uncertainties? Please cite studies showing this much uncertainty is present in the observations for a grid cell of this size (added question, what size grid cell?) and, for quantities that lack uncertainty estimates explain the reasoning behind the error distributions that were chosen.

**Author response:** We would like to stress that for the purposes of this study, our goal is not to accurately represent any particular existing sea ice observational product. Instead, we seek to characterize the general expected errors associated with each kind of observation that we assimilate, such that we can gain some understanding of their relative impact for constraining the sea ice state.

For SIC observations, we base our estimates on Zhang et al. (2018), who prescribed 15% of the true SIC value in their pan-Arctic OSSE experiments. Their use of 15% of SIC comes from Meier (2005), who performed a comparison of passive microwave SIC products from SSM/I and AVHRR imagery to

determine uncertainty in Arctic peripheral seas. Other sea ice products, such as those from EUMETSAT OSI SAF, have grid-cell level uncertainty estimates. The OSI-450 CDR, for example, has 2-3% uncertainty in open water and in 100% SIC cover, with some higher uncertainty in regions of mid-concentration, particularly at the ice-edge (Lavergne et al., 2019). Another common sea ice concentration record, the NSIDC Nasa Team estimate from NIMBUS-7 and SSM/I-SSMIS passive microwave measurements is found to have uncertainty of ~5 % of SIC in winter, and ~15% during summer (when melt ponds are present); uncertainty is lower within the ice pack, when SIC is generally high and ice is generally thicker (DiGirolamo et al., 2022). An additional study notes that "passive microwave SIC estimates in regions of consolidated ice have typically smaller uncertainties (2% to 8% SIC) than estimates from low to intermediate SIC areas with uncertainties in the order of 20% SIC…" (Wernecke et al., 2022). In response to these findings, we chose to prescribe SIC uncertainty as a parabolic function of the true SIC, which captures most of the noted dependencies on sea ice cover. The scaling factor of -0.5 was chosen such that the annual average SIC observational uncertainty in our study is 10-15%. We also note in the manuscript that the assimilation of SIC observations with $1/10^{th}$ of this standard prescribed error do constrain modeled SIC closer to TRUTH, but do little to constrain modeled SIT or SND  (Line 238).

For SIT observations, there is substantially less research to provide guidance. Early estimates of SIT from Envisat and ICESat do not come with specific uncertainty estimates at all. Later estimates from Cyrosat-2 are better constrained, as are products developed using those estimates (e.g. CS2-SMOS). The uncertainties associated with these measurements tend to be larger than those that we've prescribed (Xie et al., 2018), but SIT retrieval from remote sensing is a rapidly advancing field, and we are operating under the assumption that more accurate products (including extant ones such as those from ICESat-2 retrievals, Petty et al., 2023) are on the horizon (e.g. CRISTAL, Kern et al., 2020). Zhang et al. (2018) prescribed 0.1m everywhere as SIT uncertainty. We note from ICESat-2 thickness and freeboard measurements, there seems to be a consistent and relatively linear relationship between SIT and SIT uncertainty, with a slope that typically falls between 0.05 and 0.75 thickness estimates in pack ice. We chose 0.1 in anticipation of more accurate SIT products in the future; we also note that our annual average SIT uncertainty estimates are 4x greater than those prescribed by Zhang et al. (2018), and that expressing uncertainty as a function of thickness is a meaningful step forward from earlier work.

Finally, assimilating categorized observations is a very new idea, based for this work in the density of estimates from ICESat-2. Williams et al. (2022) attempted assimilations of categorized observations derived from real thickness measurements from Cyrosat-2 and calculated categorized uncertainties via an error analysis of concentration and thickness estimates. Under the assumption that the categorized observations could be derived by aggregating the density of observations from ICESat-2

within a model grid cell into an ice thickness distribution, we calculate categorized observation errors as standard error of the mean (SEM) of all the measurements in a grid cell. In this idealized application, we used the SEM formula for a uniform distribution (Table 3), where the bounds of the distribution are the left and right edges of the category, and we assume all points would be uniformly distributed. Our calculated uncertainties are likely underestimates of true categorized area observations, but a reasonable first step in approximating such errors.

Lines 157-158: If this is a known issue with atmospheric perturbations then why not perturb other forcings? A perturbation in the ice dynamics forcing, for example could produce a very dispersive ensemble. Please discuss the choices about perturbations and their impacts on the findings in the discussion.

**Author response:** We have experimented in other work with perturbing sea ice parameters to generate ensemble spread but find that they often lead to increases in spread that may strain realistic natural variability. By sampling the atmosphere from an ensemble reanalysis product, we ensure that the spread (and the sampling of natural variability) in the sea ice model due to variability in the forcings has some physical justification. Perturbation of the ice dynamics forcing is an idea worth considering and testing, but given the success of our present methods, we classify such experiments as outside the scope of this study. Perturbing the dynamics forcing is also an approach that would be particular to CICE-SCM-DART, and not generalizable to assimilation in the full CICE5 sea ice model. We recommend that future work examine the efficacy of including perturbations to the dynamic forcing in CICE-SCM-DART.

Line 205: If this is a standard statistical technique for this application please cite. Otherwise include a brief justification of why this statistical approach was taken.

**Author response:** Metrics such as MAE and RMSE are commonly used in OSSEs to evaluate how much assimilating observations reduces error in the model's estimate of a quantity relative to the true state. We choose to add the Welch's t-test to understand whether the reductions in error relative to TRUTH correspond to significant differences in the unassimilated, assimilated, and true timeseries. A successful experiment could then be thought of as one that is significantly different from the FREE ensemble mean and not significantly different from the TRUTH. This application also provides metric for whether one assimilation approach is meaningfully more effective than another. To clarify, the text

> *"Statistically significant differences between assimilation experiments, FREE, and TRUTH are diagnosed using a Welch's t-test."*

has been replaced with

> "In order to understand whether a) assimilating with different methods and different variables leads to meaningful adjustments toward TRUTH and b) any combinations of observations and filters significantly outperform the others over the course of the year, statistically significant differences between the ensemble mean timeseries of each EXP, FREE, and TRUTH are diagnosed using a Welch's t-test."

in Lines 222-225.

Line 210: To assess the impacts of these results, it is critical that the prescribed observational uncertainties match our actual observational uncertainties. It is not surprising that assimilating thickness is more informative than concentration for a case with thick (2m average) ice and very high ice concentration. However, our ability to measure thickness is much, much worse than concentration. Just by eye, it looks to me like the observational uncertainties prescribed in Figure 2 are overestimated for concentration and underestimated for thickness. At least for the winter, I would expect observational error in sea ice concentration in the pack ice on a 100 km grid cell to be around 0.03. In Figure 2 it looks closer to 0.1. For thickness, the summertime uncertainty should be dramatically more than the wintertime whether one is using laser or radar altimeters.

**Author response:** We agree that for the purposes of understanding the impact of a particular set of extant observations, accurately capturing the associated uncertainties is key. However, the primary goals of this study are not to evaluate the impact of any real observing system, but to 1) understanding the relative impact of assimilating different kinds of observations (to which the question of the nature of the uncertainties is indeed very important); and 2) to evaluate a series of data assimilation algorithms (to which observational uncertainties are important but perhaps less pressing). For a further discussion of the choices involved in our definitions of observational uncertainties, we refer to our response to the referee's previous comment about observational uncertainties (associated with Lines 137-142 of the original manuscript).

Line 217: The statement reads like this is a success, but shouldn't the results lie closer to the "TRUTH"? If the results were the same as FREE isn't that just an indication that the assimilation does not increase skill? Please clarify.

**Author response:** The referee is correct both that this sentence is framed as a kind of success and that the assimilation of SIC observations does not improve the analysis relative to FREE. To clarify, we found

that when SIC observations are assimilated using unbounded algorithms, they can degrade the analysis relative to FREE, because their observation error distributions are not unbounded. Because we took care to use bounded OEDs that reflect the real-world likelihood of encountering observations near sea ice bounds (i.e. that do not allow non-physical values), the unbounded filtering algorithm sees a likelihood that is biased away from the TRUTH. The use of bounded filtering algorithms accounts for this and prevents the introduction of bias into the analysis related to the OED. In that sense, the use of bounded algorithms is successful; however, the efficacy of SIC observations to adjust the analysis closer to the TRUTH is still limited. We have added a sentence, "*From this we conclude that while a bounded filter does not overcome the limited efficacy of assimilating SIC observations, respecting boundedness in the assimilation does prevent the introduction of additional bias related to assumptions about the OED*" to Lines 236-238.

Figure 4: Should the caption read "assimilating with **bounded** algorithms"?

**Author response:** Thank you for catching this! The figure title (Figure 5 in the revised manuscript) has been updated to read "*Assimilating with bounded algorithms.*"

Line 223: I assume that by "implicitly" you are referring to the aicen assimilation, but please spell that out if so.

**Author response:** As requested by the referee, we have added a clarification to this sentence, *"...that either explicitly or implicitly (through categorization in the ice thickness distribution) contains information..."* in Line 244.

Lines 224-229: If they are not statistically significantly different, then the preceding discussion of insignificant differences is not needed.

**Author response:** We agree with the referee that discussing the differences between assimilating SIT and category observations given their insignificant difference does not add to the manuscript. We have removed the sentences,

> *"Assimilating categorized observations tends to outperform assimilating SIT in terms of pMAE but falls slightly short according to iCE. This discrepancy across metrics implies that assimilating categorized observations may not capture as much of the observed variance in SIT as assimilating SIT observations, a conclusion supported by the relative pRMSE achieved in each case (Fig. A1)."*

from the revised manuscript.

Lines 231: Cite the other studies that have found this.

**Author response:** References to Blockley & Peterson (2018), Kimmeritz et al. (2018), Mu et al. (2018), Zhang et al. (2018), Fiedler et al. (2022), and Williams et al. (2022) have been added to Lines 249-250 in the revised manuscript.

Line 280: restriction on ridging should be described in the methods.

**Author response:** The ridging restriction is performed by removing a call to the dynamics forcing during the model integration. Without this call, Icepack attempts to calculate its own opening and closing rates from the ocean initial conditions for ocean $u$ and $v$, which have been set to zero. Thus, any dynamically forced ridging is negligible in those simulations. The overall effect is to prevent much very thick ice from forming in the thickest categories of the ITD. For simplicity, and because much of the ridging discussion is localized to this section, we have replaced the lines,

> *"By comparison, Fig. 9 (bottom row) presents a comparison case in which we have restricted CICE-SCM's ability to mechanically ridge ice, thereby preventing the buildup of ice in the thickest two ice categories."*

with

> *"By comparison, Fig. 10 (bottom row) presents a case in which the dynamics forcing is withheld from the model integration, thereby preventing the buildup of ice in the thickest two ice categories via mechanical processes (i.e. ridging). In all other respects, the model configuration is identical to previous experiments."*

in Lines 301-303.

Lines 280-286: The ridging turned off results took me by surprise and it's unclear to me what we are learning from them. If they are critical to the overall findings, please expand the methods (how was ridging turned off), results (what other changes did it cause in the ice state, I assume they are large), and discussion. In its current form I would recommend removing the ridging turned off section entirely.

**Author response:** The restricted ridging results emphasize the ability of categorized observations to constrain the ice state *even when there is very little ice* in some categories of the ITD. This is important

for two reasons, which we have attempted to draw out in Section 3.2. First, SIT observations begin to fail to constrain the low-ice categories, which leads to small errors in the ITD that can compound and lead to more substantial errors in the model's estimate of aggregate SIC (in this case, an underestimation of SIC compared to the TRUTH that was not noted in the thicker ice state). Assimilating categorized observations avoids this. Second, these results imply a more parsimonious solution for constraining edge cases such as:

a) very thin ice or the ice edge, where the ITD is generally skewed toward thinner categories and SIT observations have been shown to be less effective (Riedel & Anderson, 2023)

b) very thick ice, where the ITD is skewed toward thicker categories and SIC observations do a poor job adjusting SIT estimates.

This may also be relevant for reconstruction and forecasting efforts, as outlined in the manuscript. First, until very recently, SIT observations have generally been unavailable in the summertime, so a reconstructed product constrained by observations would only be able to rely on SIT observation between October and May, roughly. In that scenario, small errors in adjustments to the ITD because of SIT assimilation prior to the summer may propagate during the summer months, when only SIC observations are available to constrain the state. If those errors were minimized by assimilating categorized observations prior to the summertime observation gap, some amount of forecast error might be avoided.

We have revised phrasing throughout Section 3.2 to move away from a process-based interpretation of our results toward one that focuses on the outcomes as a function of the "balance" of the ITD.

Lines 293-294: Where were the simulations done in first-year ice regimes? Simulations with no dynamics forcing (I presume that's how ridging is turned off?) are **not** the same as first-year ice regimes, even if the annual mean thickness is similar.

**Author response:** Related to our resonse immediately prior, our intention has been to demonstrate that some combinations of observations and DA methods can constrain the sea ice state regardless of whether or not the entire ITD is represented. In doing so, we assumed that a real-world corollary for our thin ice case could be a regime that is primarily first-year ice. We thank the referee for challenging this assumption and acknowledge it's limitations. For accuracy and clarity, we have revised Section 3.2 to establish that we are not trying to simulate first-year ice regimes explicitly, but rather extrapolate how our data assimilation results might apply under different conditions. A section of planned future work will attempt to model first-year and multi-year conditions more explicitly.

Line 296: This is an overly broad statement to make from a single grid cell of multi-year ice, with a single ice dynamics and ocean forcing. Please qualify.

**Author response:** We have altered the opening sentence of Section 4 from,

> *"This study confirms that assimilating SIT observations improves sea ice analyses over assimilating SIC observations at the grid cell level."*

to

> *"This work reinforces the results of previous studies that assimilating SIT observations generally improves sea ice analyses over assimilating SIC observations alone."*

in Lines 319-320. We also note the subsequent sentence, which qualifies our results to "*this ensemble and mean state*" (Lines 320-322).

Line 305: I do not see the thin ice simulations that this is referring to. If this is a critical finding from this work, it needs to be supported by simulations of seasonal ice. This should be straightforward to accomplish with the existing code infrastructure (i.e., pick a more southerly point for the atmospheric and oceanic forcing), but would entail significant additional work. Alternatively, the findings could be rewritten to describe just the presented work and simulations in seasonal ice conditions could be recommended as future work.

**Author response:** The inclusion of the referee's suggested bug fix related to the dynamics forcing implementation in Icepack (comment referring to Line 92 of the original manuscript) produces ensembles that better reflect the seasonal nature of the location from which our forcings are drawn. The ensemble is thinner overall and more seasonal, with several members that drop to zero SIC in the summer months. We have rerun all our assimilation experiments with this bug fix implemented and updated all the figures in the paper with the results. The inclusion of the bug fix and the thinner ice state result in small quantitative changes, but do not alter the qualitative conclusions of the study (which should be expected, given that both ensemble spread and observation uncertainty tend to decline under a thinner mean state). Previously mentioned future work will attempt to simulate a wider range of the Arctic's sea ice states, including coastal ice, Central Arctic pack ice, and very seasonal ice in the Barents Sea.

Line 332: Is this really needed? The aicen lines on the figure seem to overlap with the "TRUTH" almost entirely. At what point is a DA system good enough?

**Author response:** We thank the referee for this question—it is an observant one, particularly as it pertains to how good a DA system needs to be. The latter is an open question that we believe is somewhat dependent upon the application for that DA system. An approach to constraining the categories of the ITD *and* aggregate SIC simultaneously is, in our view, still necessary. First, as illustrated in Figure 11, even the DA algorithms that respect bounds and are used to assimilate categorized observations occasionally require SIC postprocessing. This is because within the current framework the assimilation can respect the categorized area bounds (which are necessarily 0 and 1 for every category), but not the aggregate of the area categories. A solution that can remove the need to such manual postprocessing entirely would be ideal. Second, as previously noted in our discussion of observational uncertainties, it is likely that we have underestimated the uncertainty associated with each categorized observation. As such, it is possible that categorized assimilation may be less accurate than presented here (though still effective, due to the linearity between category observations and the model state variables).

Line 334: It would be helpful to the field to expand the description of what kind of targeted covariance study is needed. Based on your findings, what should field scientists, remote sensing researchers, and modelers do next?

**Author response:** We are actively engaged in research that explores the covariance relationship in observations between sea ice concentration and sea ice thickness. For sea ice DA, we list a few key points that we believe would helpful foci:

a) For the purposes of understanding variability that the ensemble seeks to sample: what is the natural variability of SIC and SIT on a local/grid-cell scale? How does this variability change across the Arctic?

b) For the purposes of understanding how observable variables relate to one another and to state space variables: What are typical covariance values on the local/grid-cell level between SIC and SIT? How do aggregate observable covary with each category of the ITD? What are the length scales of those covariance relationships? How well do models capture those covariance relationships compared to observations?

We have added two sentences to Section 5 that suggest a covariance study for local SIC and SIT (Lines 357-359).

**References**

Anderson, J. L.: An ensemble adjustment kalman filter for data assimilation, Mon. Wea. Rev., 129, 2884– 2903, https://doi.org/10.1175/1520- 0493(2001)129<2884:AEAKFF>2.0.CO;2, 2001.

Brennan, M. K., and G. J. Hakim: Reconstructing Arctic Sea Ice over the Common Era Using Data Assimilation, J. Climate, 35, 1231–1247, https://doi.org/10.1175/JCLI-D-21-0099.1, 2022.

DiGirolamo, N. E., C. L. Parkinson, D. J. Cavalieri, P. Gloersen, and H. J. Zwally: Sea Ice Concentrations from Nimbus-7 SMMR and DMSP SSM/I-SSMIS Passive Microwave Data, Version 2 User Guide, NASA National Snow and Ice Data Center Distributed Active Archive Center, https://doi.org/10.5067/MPYG15WAA4WX, 2022.

Evensen, G.: The ensemble Kalman filter: Theoretical formulation and practical implementation, Ocean Dyn., 53, 343-367, 2003.

Kern, M., Cullen, R., Berruti, B., Bouffard, J., Casal, T., Drinkwater, M. R., Gabriele, A., Lecuyot, A., Ludwig, M., Midthassel, R., Navas Traver, I., Parrinello, T., Ressler, G., Andersson, E., Martin-Puig, C., Andersen, O., Bartsch, A., Farrell, S., Fleury, S., Gascoin, S., Guillot, A., Humbert, A., Rinne, E., Shepherd, A., van den Broeke, M. R., and Yackel, J.: The Copernicus Polar Ice and Snow Topography Altimeter (CRISTAL) high-priority candidate mission, The Cryosphere, 14, 2235– 2251, https://doi.org/10.5194/tc-14-2235-2020, 2020.

Landy, J.C., Dawson, G.J., Tsamados, M. et al.: A year-round satellite sea-ice thickness record from CryoSat-2, Nature, 609, 517–522, https://doi.org/10.1038/s41586-022-05058-5, 2022.

Lavergne, T., Sørensen, A. M., Kern, S., Tonboe, R., Notz, D., Aaboe, S., Bell, L., Dybkjær, G., Eastwood, S., Gabarro, C., Heygster, G., Killie, M. A., Brandt Kreiner, M., Lavelle, J., Saldo, R., Sandven, S., and Pedersen, L. T.: Version 2 of the EUMETSAT OSI SAF and ESA CCI sea-ice concentration climate data records, The Cryosphere, 13, 49–78, https://doi.org/10.5194/tc-13-49-2019, 2019.

Meier, W. N.: Comparison of passive microwave ice concentration algorithm retrievals with AVHRR imagery in arctic peripheral seas, IEEE Transactions on Geoscience and Remote Sensing, 6, 1324-37, https://doi.org/10.1109/TGRS.2005.846151, 2005.

Petty, A. A., Keeney, N., Cabaj, A., Kushner, P., and Bagnardi, M.: Winter Arctic sea ice thickness from ICESat-2: upgrades to freeboard and snow loading estimates and an assessment of the first three winters of data collection, The Cryosphere, 17, 127–156, https://doi.org/10.5194/tc-17-127-2023, 2023.

Riedel, C. and Anderson, J.: Exploring Non-Gaussian Sea Ice Characteristics via Observing System Simulation Experiments, EGUsphere [preprint], https://doi.org/10.5194/egusphere-2023-96, 2023.

Wernecke, A., Notz, D., Kern, S., and Lavergne, T.: Estimating the uncertainty of sea-ice area and sea-ice extent from satellite retrievals, EGUsphere [preprint], https://doi.org/10.5194/egusphere-2022-1189, 2022.

Williams, N., Byrne, N., Feltham, D., Van Leeuwen, P. J., Bannister, R., Schroeder, D., Ridout, A., and Nerger, L.: The effects of assimilating a sub-grid-scale sea ice thickness distribution in a new Arctic sea ice data assimilation system, The Cryosphere, 17, 2509–2532, https://doi.org/10.5194/tc-17-2509-2023, 2023.

Xie, J., Counillon, F., and Bertino, L.: Impact of assimilating a merged sea-ice thickness from CryoSat-2 and SMOS in the Arctic reanalysis, The Cryosphere, 12, 3671–3691, https://doi.org/10.5194/tc-12-3671-2018, 2018.

Zhang, Y., C. M. Bitz, J. L. Anderson, N. Collins, J. Hendricks, T. Hoar, K. Raeder, and F. Massonnet: Insights on Sea Ice Data Assimilation from Perfect Model Observing System Simulation Experiments, J. Climate, 31, 5911–5926, https://doi.org/10.1175/JCLI-D-17-0904.1, 2018.

---

## Author Comment (AC2)

**Bounded & categorized: targeting data assimilation for sea ice fractional coverage and non-negative quantities in a single column multi-category sea ice model**

*Manuscript egusphere-2023-2016*

**Responses to Referee 2**

The authors would like to thank the editor and Referee 2 for the time and effort that have gone toward providing feedback on this manuscript. Please find below, in blue, our responses to referee comments, questions, and concerns. All page numbers refer to the revised manuscript.

*General Comments*

The authors present a framework for hypothesis testing in sea ice data assimilation (DA). Sea ice DA is complicated by the bounds on the sea ice variables, i.e., SIC and SIT should be greater than zero and SIC should be less than or equal to one. The single column sea ice model Icepack and the Data Assimilation Research Testbed (DART) are used. Non-Gaussian error covariances are tested for SIC, SIT, and category-based assimilation.

The paper is well written, relatively easy to understand (given the relatively complicated topic), and definitely deserves publication. I thank the authors for this nice work! I enjoyed reading it.

However, any paper can be improved. Below I have listed a few points that I would like to address. Overall, I rate them as minor revisions.

**Author response:** We thank Referee 2 for their kind feedback and are glad to hear the paper was an enjoyable read! Our responses to individual comments are included below.

*Detailed Comments*

I miss a statement in the introduction (and the abstract) that no 'real' observations are used, but perfect model studies, i.e. Observing System Simulation Experiments, are performed. This should be made clear from the start.

**Author response:** We are grateful to the referee for highlighting this omission! Clarification that the simulations performed for this study are a type of OSSE is a valuable addition to the manuscript that avoids additional confusion about the synthetic nature of the observations. We have added phrasing to the Abstract (Lines 5-7) and to the Section 2 introduction (Lines 86-89) that clarify our motivation for (and use of) the OSSE framework.

Line 76: I would add that DART will be explained in section 2.2.

**Author response:** References to further discussion of Icepack in section 2.1 and DART in section 2.2 have been added to the manuscript (Lines 83-84).

Line 77: To call Icepack a "single column version of the CICE sea ice model" sounds a bit strange, because the Icepack documentation says "The column physics package of the CICE sea ice model, 'Icepack' …", i.e. Icepack is not a specific version of CICE, but an integral part of CICE. That might cause confusing as well as calling the data assimilation framework "CICE-SCM-DART".

**Author response:** We thank the referee for highlighting the potential for confusion here. We have altered the introduction of Icepack from

> *"The data assimilation framework used in this study couples the Data Assimilation Research Testbed (DART, Anderson et al., 2009) to Icepack (Icepack, 2020), a single-column version of the CICE sea ice model; the latter is widely used as the sea ice component of several Earth system models and in stand-alone sea ice studies. In keeping with naming conventions developed in coincident work (Riedel et al., 2023), this framework is referred to as CICE-SCM-DART."*

to

> *"The data assimilation framework used in this study couples the Data Assimilation Research Testbed (DART, Anderson et al., 2009) to Icepack (version 1.3.1, Icepack, 2020), the column-physics package of the CICE sea ice model, which is widely used as the sea ice component of several Earth system models. Icepack can be run in a stand-alone configuration as a sort of single-column model and is reviewed in Section 2.1. DART is discussed in more depth in Section 2.2. In keeping with naming conventions developed in coincident work (Riedel et al., 2023), the collective assimilation system is referred to as CICE-SCM-DART."*

in Lines 80-85.

We have retained the use of CICE-SCM-DART as a name for the framework to be consistent with other work (Riedel et al., 2023) and to reflect our use case of Icepack in its standalone setting.

Line 88: Explain briefly what CAM6 is.

**Author response:** In Line 88 of the original manuscript, we are referring to an ensemble reconstruction product that was run in CAM6 using DART, not to CAM6 itself. To avoid unnecessary detail or confusion, we have revised this sentence from,

> *"For use in the CICE-SCM-DART framework, 30 instances of Icepack are forced by unique atmospheric conditions extracted from randomly selected members of the CAM6 + DART reanalysis product (Raeder et al., 2021)."*

to

> *"For use in the CICE-SCM-DART framework, 30 instances of Icepack are forced by unique atmospheric conditions extracted from randomly selected members of a recent large-ensemble reanalysis product (Raeder et al., 2021)."*

in Lines 93-94.

A later reference to CAM6 (in relation to models DART can be integrated with) is qualified with explanatory phrasing in Line 116,

> *"... the Community Atmosphere Model (CAM6), the atmosphere component of the CESM2 climate model."*

Line 89: It would help the reader if a little bit more was said about the consequences of "setting the snow grain size parameter to a value of -2". Why does this choice "prevents discontinuous behavior in ice concentration related to ice-albedo feedback during the melt season"?

**Author response:** Please see our response to a similar comment from Referee 1 (related to Line 90 of the original manuscript).

Line 93: What does "are consistent" mean? I assume it means that the values are the same for all 30 members, right?

**Author response:** The referee is correct—the ocean forcing values are the same for all 30 members of the sea ice ensemble. For clarity, "*consistent*" has been replaced with "*identical*" in the revised manuscript (Line 103).

Line 126: "The use of bounded normal rank histogram (BNRH) distributions in state-space regression is addressed in (Anderson, 2023)": I would prefer to read here a few sentences about the main findings of Anderson (2023).

**Author response:** We have added sentences to this effect to the end of Section 2.2 (Lines 138-141). The paragraph now concludes,

> *"The use of bounded normal rank histogram (BNRH) distributions in state-space regression (step 4) of the QCEFF enforces appropriate bounds by way of a series of transforms in probit and probability integral space. This aspect of the QCEFF also more deftly handles nonlinear relationships between observed quantities and modeled state variables and is addressed in depth for idealized cases in Anderson (2023)."*

Line 145: Table 2 is repeating many information four times (i.e. obs. Kind and obs. Error). I suggest to split the table into two – one naming the experiments and the other describing the obs. Error (which is the same for all experiments).

**Author response:** Per the referee suggestion, we have split Table 2 into Table 2 and Table 3. The former lists the experiments, and the latter outlines the associated observation uncertainty for each kind of observation.

Table 2: The obs. Error of SIT (10% of SIT value) is unrealistic low at least when compared to obs. Errors from altimetry (see e.g. Figure 2b in https://tc.copernicus.org/articles/11/1607/2017/tc-11-1607-2017.pdf). Any comment on that?

**Author response:** For a more in-depth discussion of the reasoning behind our definitions for observational uncertainties, including for SIT, please refer to our previous response to a similar comment from Referee 1 (related to Lines 137-142 of the original manuscript). Briefly, our SIT uncertainties are motivated by the anticipation of both improvements to SIT retrieval methods from current products (Landy et al., 2022) and the likelihood of increasingly accurate SIT measurements from current and future observing missions such as ICESat-2 (Petty et al., 2023) and CRISTAL (Kern et al., 2020).

Line 166: I found the sentence "As a result, an observation from any of individual ITD categories is prevented from updating any state-space variable not also in that same ITD category." difficult to understand. A reformulation of the sentence helped me: "As a result, an observation from any of the

individual ITD categories is prevented from updating any state-space variable that is not also in the same ITD category".

**Author response:** We thank the referee for this suggestion! We agree that the proposed reformulation is clearer and have replaced the sentence accordingly (Lines 181-182 in the revised manuscript).

Line 174: "CICE rebalancing option" – please explain briefly what that is!

**Author response:** We have included some clarification around the CICE rebalancing option for postprocessing.

> *"All experiments in Table 2 make use of the CICE rebalancing option."*

now reads,

> *"All experiments in Table 2 make use of this default rebalancing option, which redistributes the ice fractional coverage in each category to ensure that the thickness bounds are respected and then calculates consistent ice and snow volumes, salinities, and enthalpies once the updates have occurred."*

in Lines 188-190.

Line 190: Is NSE not more commonly used as CE? - see
https://en.wikipedia.org/wiki/Nash%E2%80%93Sutcliffe_model_efficiency_coefficient

**Author response:** NSE and CE are definitionally equivalent. CE is a term for the metric commonly used in paleoclimate applications of DA (Klein & Goosse, 2017; Steiger & Hakim, 2014; Brennan & Hakim 2022); most studies using CE in this context cite Nash & Sutcliffe (1970).

Line 204: iCE is to me not more intuitive!

**Author response:** We have rephrased this sentence to avoid implications of how a reader should engage with iCE (Line 220). We retain the use of iCE because the idealized nature of our experiments means that most of our simulations have a high CE to begin with (i.e. the model with data assimilation is a good predictor of TRUTH); to get a more nuanced picture of how different observation kinds or DA algorithms influence CE, we present the CE improvement (iCE) for each experiment relative to the baseline CE of the FREE case as a predictor of the TRUTH.

Figure 4: I question the usefulness of discussing the snow depth variable in the manuscript. To me it is just a way of diluting the results without learning anything essential.

**Author response:** This is a valid point, and we thank the referee for raising it. We choose to discuss the snow results in this context for 3 reasons:

a) Snow atop the sea ice is an important part of the sea ice state in the Arctic and plays a critical role in melt and freeze-up processes. We have seen this in our own work with respect the influence of changing snow parameters in the model (see earlier response to comments about $R\_snw$).

b) Snow is a key variable in the derivation of sea ice freeboard, which precedes sea ice thickness calculations in most retrieval algorithms. Snow loading estimates in the Arctic are sparse or depend heavily on models however, and there has some idea that assimilating sea ice observations might help constrain snow estimates more accurately. We've shown that this is not likely to be the case, though there may be some small improvement when assimilating category observations (Figure 6, bottom left). This finding emphasizes a need for better snow observations.

c) Coincident work (Riedel et al., 2023) explores the impact of assimilating SIT, SIC, and SND observations in a perfect model OSSE context in the CICE-SCM-DART framework. The inclusion of the snow results here provides some nice additional context for that work.

Line 265: ",the dependency". White space is missing!

**Author response:** The whitespace has been added (Lines 286-287).

Line 290: It is true that SIT summer observations were missing (at least space filling) , but I would point to the forthcoming data products (e.g. https://www.nature.com/articles/s41586-022-05058-5).

**Author response:** We thank the referee for highlighting recent advances in SIT retrievals. The potential to have and to assimilated summertime SIT measurements is an exciting prospect, and one we are eager to explore in future work! However, we also anticipate that there may be applications for sea ice DA outside the 2011-2021 period for which the Landy et al. (2022) product is currently available, as well as for purposes of comparison and validation.

**References**

Brennan, M. K., and G. J. Hakim: Reconstructing Arctic Sea Ice over the Common Era Using Data Assimilation, J. Climate, 35, 1231–1247, https://doi.org/10.1175/JCLI-D-21-0099.1, 2022.

Klein, F., Goosse, H.: Reconstructing East African rainfall and Indian Ocean sea surface temperatures over the last centuries using data assimilation, Clim Dyn 50, 3909–3929, https://doi.org/10.1007/s00382-017-3853-0, 2018.

Landy, J.C., Dawson, G.J., Tsamados, M. et al.: A year-round satellite sea-ice thickness record from CryoSat-2, Nature, 609, 517–522, https://doi.org/10.1038/s41586-022-05058-5, 2022.

Nash, J.E. and Sutcliffe, J.V.: River flow forecasting through conceptual models, Part I: a discussion of principles, J. Hydrol., 10, 282-290, https://doi.org/10.1016/0022-1694(70)90255-6, 1970.

Petty, A. A., Keeney, N., Cabaj, A., Kushner, P., and Bagnardi, M.: Winter Arctic sea ice thickness from ICESat-2: upgrades to freeboard and snow loading estimates and an assessment of the first three winters of data collection, The Cryosphere, 17, 127–156, https://doi.org/10.5194/tc-17-127-2023, 2023.

Riedel, C., Wieringa, M., and Anderson, J.: Exploring Bounded Non-parametric Ensemble Filter Impacts on Sea Ice Data Assimilation, Mon. Wea. Review [in press], (2023)

Steiger, N. and Hakim, G.: Multi-timescale data assimilation for atmosphere–ocean state estimates, Clim. Past, 12, 1375–1388, https://doi.org/10.5194/cp-12-1375-2016, 2016.